# Organic carbon densities and accumulation rates in surface sediments of the North Sea and Skagerrak

Markus Diesing[1], Terje Thorsnes[1], Lilja Rún Bjarnadóttir[1]

[1]Geological Survey of Norway, Postal Box 6315 Torgarden, 7491 Trondheim, Norway

5   *Correspondence to*: Markus Diesing (markus.diesing@ngu.no)

**Abstract.** Continental shelf sediments are places of both rapid organic carbon turnover and accumulation, while at the same time increasingly subjected to human-induced disturbances. Recent research suggests that shelf sediments might have a role to play as a natural climate solution, e.g., by storing organic carbon if left undisturbed from anthropogenic activity. However, we have an incomplete understanding about the centres of organic carbon accumulation and storage on continental shelves. To better constrain the rate of accumulation and the mass of organic carbon that is stored in sediments, we developed and applied a spatial modelling framework that allows us to estimate those quantities from sparse observations and predictor variables known or suspected to influence the spatial patterns of these parameters. This paper presents spatial distribution patterns of organic carbon densities and accumulation rates in the North Sea and Skagerrak. We found that organic carbon stocks and accumulation rates are highest in the Norwegian Trough, while large parts of the North Sea are characterised by low stocks and zero net-accumulation. The total stock of organic carbon that is stored in the upper 0.1 m of sediments amounted to 230.5 ± 134.5 Tg C, of which approximately 26 % are stored in the Norwegian Trough. Rates of organic carbon accumulation in the Norwegian Trough are comparable with those reported from nearby fjords. We provide baseline datasets that could be used in marine management, e.g., for the establishment of "carbon protection zones". Additionally, we highlight the complex nature of continental shelves with zones of rapid carbon cycling and accumulation juxtaposed, which will require further detailed and spatially explicit analyses to constrain sedimentary organic carbon stocks and accumulation rates globally.

## 1 Introduction

Marine sediments are an important sink for organic carbon (OC) on Earth, with estimates of OC burial in marine sediments ranging from 126 Tg C yr$^{-1}$ (Berner, 1982) to 350 Tg C yr$^{-1}$ (Keil, 2017). The major hot spots for OC burial in the global ocean are the coastal margins (Bianchi et al., 2018). Burdige (2007) estimated that 80% (248 Tg C yr$^{-1}$) of all OC buried in marine sediments is occurring in continental margin sediments. However, other estimates do also exist (Bauer et al., 2013; Duarte et al., 2005; Hedges and Keil, 1995), ranging from 45.2 to 300 Tg C yr$^{-1}$, and budgets are generally not well constrained (Burdige, 2007). Estimates of the amount of OC stored in marine surface sediments also vary considerably, ranging from 87 Pg C (Lee et al., 2019) to 168 Pg C (LaRowe et al., 2020) to 3117 Pg C (Atwood et al., 2020). Such

differences can be attributed only partly to differences in the reference depths being considered, ranging from 0 – 5 cm (Lee et al., 2019) to the bioturbated Holocene layer, assumed to be 0 – 10 cm (LaRowe et al., 2020), to 0 – 1 m (Atwood et al., 2020).

In recent years, attempts have been made to construct carbon budgets for entire continental shelf systems. However, these studies have not included spatially explicit estimates of OC stock and burial (Fennel et al., 2019; Najjar et al., 2018) or

concluded that both stocks and burial rates were associated with considerable uncertainty (Legge et al., 2020).

Given the importance of continental margins in OC cycling, it is therefore of great importance to develop adequate methods that better constrain stocks, flows and budgets of OC and quantify the uncertainty of the predictions. In particular, spatially explicit methods that predict the variation of OC in space by means of geostatistics or machine-learning spatial prediction are promising, and much can be learned from related terrestrial disciplines such as digital soil mapping (Hengl et al., 2014,

2017; McBratney et al., 2003). Recent studies appear to prefer machine-learning over geostatistical approaches (Seiter et al., 2004) due to their performance, flexibility, and generality (Hengl et al., 2018), and estimates of OC stored in marine sediments at a global (Atwood et al., 2020; Lee et al., 2019) and sea-basin scale (Diesing et al., 2017; Wilson et al., 2018) have been derived. However, no spatially explicit estimates of OC accumulation and burial rates exist to our knowledge.

It is important to stress the difference between OC burial and OC accumulation here. Burial is the deposition of OC below

the zone of active degradation (Keil, 2015). OC degradation in surficial seafloor sediments happens via various processes including aerobic respiration, denitrification, manganese reduction, iron reduction, sulfate reduction and methanogenesis (Berner, 1980). Burial thus is the removal of OC from the active carbon cycle and the burial rate can be expressed as the product of sediment accumulation and OC content at the depth below which no further degradation of OC occurs (Middelburg, 2019). It is, however, difficult to determine that depth. Various depth horizons have been used, e.g. the lower

boundary of the sulfate reduction zone (Jørgensen et al., 1990), 15 cm (Hartnett et al., 1998) and 10 cm (Bakker and Helder, 1993). OC accumulation rates, however, can be calculated for any specific depth interval of the sediment column. Due to the difficulties of determining the relevant depth to estimate burial rates and the scarcity of burial rate data, we decided to estimate OC accumulation rates instead.

Well-constrained estimates of OC stocks and accumulation rates are also required from a marine management perspective.

OC stocks are a measure of the vulnerability potential, while accumulation rates are a measure of the mitigation potential (Jennerjahn, 2020). The potential of so-called Blue Carbon ecosystems (mangroves, salt marshes, seagrass meadows and potentially macroalgae (Krause-Jensen and Duarte, 2016)) to sequester and store OC is an important ecosystem service that has been highlighted in recent years (Duarte et al., 2005; Mcleod et al., 2011; Nellemann et al., 2009). More recently, it has been shown that fjord (Smeaton et al., 2016, 2017) and continental shelf sediments (Diesing et al., 2017) harbour

considerable amounts of OC. In the United Kingdom, the shelf sediment stock (205 Tg C) accounts for 93% of OC stored in coastal and marine habitats (Luisetti et al., 2019) and outweighs combined seagrass and saltmarsh stocks (13.4 Tg C) by a factor of ≈15. In Namibia, the marine sediment OC stock is estimated to be larger than the soil OC stock (Avelar et al., 2017). Determining national carbon stocks is essential to understand the potential vulnerability of those stocks to human

activities; however, national assessments for greenhouse gas reporting do not account for marine stocks such as organic carbon stored in shelf sediments (Avelar et al., 2017). In Norway, the government has underlined the significance of OC uptake by marine vegetation but OC accumulation in marine sediments is currently not considered (Anon, 2013). Consequently, the question has been raised whether those stocks should be considered as part of national carbon accounting and potential greenhouse gas mitigation strategies and subject to management against human-induced disturbance (Avelar et al., 2017). The socio-economic importance of marine carbon storage has recently been assessed in a scenario analysis of increased human and climate pressures over a 25-year period. It was estimated that damage costs of up to $12.5 billion from carbon release linked to disturbance of coastal (areal loss of seagrass habitats, sediment OC loss from saltmarshes) and shelf sea sediment (resuspension by bottom contact fishing) carbon stores could arise in the United Kingdom (Luisetti et al., 2019). However, the transboundary nature of carbon flows in the marine environment poses significant challenges for carbon accounting and requires new guidance and governance frameworks to manage these stocks (Luisetti et al., 2020).

Marine Protected Areas (MPAs) might be a suitable management measure to effectively protect the carbon storage ecosystem service of Blue Carbon ecosystems against human pressures (Zarate-Barrera and Maldonado, 2015) by slowing, halting, or reversing the trend of degradation and loss of e.g. seagrass and mangrove ecosystems. In Indonesia, MPAs reduced mangrove loss by about 140 km$^2$ and avoided emissions of 13 Tg $CO_2$ equivalent between 2000 and 2010 (Miteva et al., 2015). Further offshore, demersal fishing is an important and widespread pressure on continental shelf seabed habitats (Amoroso et al., 2018; Halpern et al., 2008). Chronic demersal fishing has negative impacts on benthic biomass, production, and species richness, and is leading to shifts in the composition of communities (Hiddink et al., 2006, 2017; Jennings et al., 2001; Tillin et al., 2006). The impact of demersal fishing on the biogeochemistry of the seafloor and OC storage is less well understood. Several studies show lower OC contents in surface sediments of trawled areas (Bhagirathan et al., 2010; Martín et al., 2014b; Paradis et al., 2019, 2020; Pusceddu et al., 2014), while others report higher OC contents, presumably due to fertilization brought about by resuspension or uplifting of OC from deeper layers caused by trawling (Palanques et al., 2014; Pusceddu et al., 2005). In the short term, demersal fishing-induced sediment disturbance stimulates OC mineralisation in cohesive sediments, likely due to the enhanced decomposition of previously buried refractory OC (van de Velde et al., 2018). In the long-term, the expected result of repeated and vigorous sediment mixing due to demersal fishing is a general impoverishment in OC (Martín et al., 2014a). Given the large areas affected (≈10 million km$^2$) and the amount of sediment being resuspended (≈22 Pg C yr$^{-1}$) globally (Oberle et al., 2016), it is likely that the impact of demersal fishing on shelf sediment OC storage is substantial. Chronic seabed disturbance by demersal fishing might have a sizeable impact on the carbon cycle in cohesive sediments on continental shelves by keeping coastal seabed biogeochemistry in a transient state, which translates into reduced OC accumulation rates (van de Velde et al., 2018). Establishment of MPAs protecting against demersal fishing could not only facilitate the recovery of benthic species but also promote longer-term carbon uptake by seabed ecosystems through increased biomass, as well as prevent further loss of OC stored in sediments (Roberts et al., 2017).

The North Sea and Skagerrak are among the most intensively researched regional seas with a wealth of data available for reuse. At the same time, they are the most heavily impacted by human activities (Halpern et al., 2008). This makes the area ideal for our study which has the objectives to estimate OC stocks and accumulation rates of surface sediments in a regional sea that is impacted by human activities. These estimates will be accompanied by assessments of uncertainty in the predictions. With the help of these predictions, the following research questions will be addressed:

1. What is the importance of seafloor sediment OC stocks relative to other OC stocks?
2. Where are the centres of OC accumulation in the North Sea and Skagerrak?
3. Based on the previous results, can we differentiate between different zones of OC processing at the seafloor?
4. What are possible implications for marine management?

## 2 Regional setting

The study site encompasses the North Sea and Skagerrak regional seas as defined by IHO (1953). The surface areas of the North Sea and Skagerrak are approximately 526,000 km$^2$ and 32,000 km$^2$, respectively. The seafloor in the study site is mostly shallow and flat, generally deepening from south to north (Fig. 1). The most prominent morphological feature is the Norwegian Trough, which follows the coast of southern Norway and reaches water depths of nearly 700 m in the Skagerrak. It forms a major accumulation area for fine-grained material (Eisma and Kalf, 1987; Van Weering, 1981). Large parts of the continental shelf outside the Norwegian Trough are erosional or non-depositional in nature (de Haas et al., 1997), with limited sedimentation occurring in the German Bight, the Elbe palaeo-valley, Oyster Ground, Inner Silver Pit, Outer Silver Pit and Devil's Hole (Eisma and Kalf, 1987; de Haas et al., 1997). Previous studies (de Haas et al., 1997, 2002; de Haas and van Weering, 1997) have indicated that most of the OC accumulation occurs in the Norwegian Trough ($\approx$1 Tg C yr$^{-1}$), while OC accumulation in the remaining area is low ($\approx$0.1 Tg C yr$^{-1}$).

## 3 Data

### 3.1 Response variables

### 3.1.1 Linear sedimentation rate

Linear sedimentation rate ($\omega$), measured in (cm yr$^{-1}$), is used here synonymously with sediment accumulation rate. Data were initially sourced from the EMODnet-Geology portal (https://www.emodnet-geology.eu/), which provides a collation of values from the literature across European sea basins. The dataset was limited to the study site and sedimentation rates based on $^{210}$Pb, to ensure a consistent integration time scale (Jenkins, 2018). Based on a half-life of approximately 22 yr, the associated integration time is roughly 100 yr (Jenkins, 2018). Data from Zuo et al. (1989) were excluded as these were deemed unreliable (de Haas et al., 1997).

The reported sedimentation rate data focussed on accumulation areas like the Norwegian Trough (Figure 2). However, to be able to spatially predict sedimentation rates across the study site it is necessary to include data from areas of erosion and non-deposition, which predominate in the North Sea. Therefore, the data of de Haas et al. (1997) were also included. This provided less than 20 data points of zero net-sedimentation, which was still deemed insufficient. Additionally, pseudo-observations (Hengl et al., 2017) were also included. Pseudo-observations are 'virtual' samples that are placed in undersampled areas and for which the value of the response variable can be assumed with high certainty. Hengl et al. (2017) cite 0 % soil OC in the top 2 m of active sand dunes as an example. Mitchell et al. (2021) placed pseudo-samples in areas of bedrock outcropping at the seabed when predicting sedimentation rates in the Baltic Sea. The placement of pseudo-observations was restricted to areas of erosion and non-deposition (based on the sedimentary environment layer, as described in chapter 3.2), for which a sedimentation rate of 0 cm yr$^{-1}$ could be assumed. The pseudo-observations were placed randomly to avoid human bias. Some of the sedimentation rate values from non-depositional areas reported by de Haas et al. (1997) and van Weering et al. (1993) appeared too high, and after a review of the $^{210}$Pb-profiles four of them were set to 0 cm yr$^{-1}$ due to low $^{210}$Pb activities and indistinct decreases with depth. The full dataset used for subsequent modelling is shown in Fig. 2 and provided as Supplementary Data Table 1.

### 3.1.2 Organic carbon density

Previous studies have predicted OC content and sediment porosity separately to calculate OC stocks (Diesing et al., 2017; Lee et al., 2019; Wilson et al., 2018). Here, we first calculate OC density from concurrent measurements of OC content and sediment dry bulk densities or porosities. This has two advantages: First, there is no need to transform the response variable as would be necessary in the case of OC content reported as weight-% or fractions. Second, only one model instead of two needs to be fitted. This is advantageous as fitting two models would likely increase the uncertainty of the predictions. Initially, a wide range of data sources were accessed. Ultimately, 373 samples fulfilled the criterion of providing OC content and dry bulk density/porosity measured on the same sample. These samples were collected and measured by the Geological Survey of Norway, the Centre for Environment, Fisheries and Aquaculture Science, Bakker and Helder (1993) and de Haas et al. (1997). The full dataset used for subsequent modelling is shown in Figure 2 and provided as Supplementary Data Table 2.

OC density $\rho_{OC}$ (kg m$^{-3}$) was calculated from data on OC content G (g kg$^{-1}$) and dry bulk density $\rho_d$ (kg m$^{-3}$):

$$\rho_{OC} = G \cdot \rho_d \tag{1}$$

If not measured, dry bulk density was calculated from porosity $\varphi$ and the grain density $\rho_s$ (2650 kg m$^{-3}$) according to:

$$\rho_d = (1 - \varphi)\rho_s \tag{2}$$

In the majority of cases (52.8 %), the OC concentrations referred to the 0 – 10 cm depth interval, but other depth intervals were also present; most frequently 0 – 1 cm (17.7 %), 0 – 5 cm (16.4 %), 0 – 0.5 cm (6.7 %) and 0 – 2 cm (4.6 %). It was assumed that the reported values were representative for the upper 10 cm of the sediment column. The full dataset used for subsequent modelling is shown in Fig. 2 and provided as Supplementary Data Table 2.

## 3.2 Predictor variables

The initial selection of environmental predictor variables was based on availability and expected relevance to OC. At this initial stage of conceptual model building (Guisan and Zimmermann, 2000), it might be prudent to include a wide range of potentially relevant variables. A selection of variables that are actually relevant for the model will be performed subsequently. A previous modelling study highlighted mud content in surficial sediments, bottom water temperature and distance to the closest shoreline as important predictors for OC (Diesing et al., 2017). Other environmental controls on OC accumulation that have been inferred are sedimentation rate (Müller and Suess, 1979), bottom-water oxygen concentration (Paropkari et al., 1992) and oxygen exposure time (Hartnett et al., 1998). There is less information available on relevant predictors for sedimentation rate, but it is assumed that sedimentation is favoured in deep basins with low current speeds and wave orbital velocities. Fine grained sediments prevail in these environments and might be indicative for areas of sediment accumulation.

Some predictor variables were derived from other data layers: The geomorphology layer was derived from Harris et al. (2014) and contained the geomorphic features shelf, shelf valley and glacial trough. The sedimentary environment was inferred from modelled Folk classes (Mitchell et al., 2019). Initially, areas covered with mud, sandy mud and muddy sand were assumed to be potentially accumulative. Boundaries were subsequently cleaned in ArcGIS to simplify the regions. These potential accumulation areas were critically reviewed in the light of measured sedimentation rates and geological interpretations of sediment cores (de Haas et al., 1997 and references therein). The remaining main areas of net-deposition are shown in Fig. 1. The process is shown in Fig. A1. Oxygen penetration depth was derived by applying relationships between measured oxygen penetration depth and mud content (pers. comm. John Barry, Cefas) to the mud layer (Mitchell et al., 2019). Oxygen exposure time was derived by dividing oxygen penetration depth by the modelled linear sedimentation rate (Hartnett et al., 1998).

All datasets were projected to Lambert Azimuthal Equal Area projection with a resolution of 500 m. The full list of predictor variables is detailed in Table 1.

## 4 Methods

## 4.1 Framework for spatial prediction and uncertainty estimation

The same modelling framework was used for predicting sedimentation rates and OC densities. It is based on the quantile regression forest (QRF) algorithm (Meinshausen, 2006) to make spatial predictions of the response variables and to estimate

the uncertainty in the predictions in a spatially explicit way. QRF is a generalisation of the random forest algorithm (Breiman, 2001), which aggregates the conditional mean from each tree in a forest to make an ensemble prediction. QRF also returns the whole conditional distribution of the response variable. This allows us to determine the underlying variability of an estimate by means of prediction intervals or the standard deviation.

Prediction uncertainty may be divided into four main components: uncertainty in the response data, in the predictor
variables, in the model and in variations of available data (Guevara et al., 2018). It was not possible to address uncertainty related to the first two components, as information on measurement error of the response variables or uncertainty associated with the predictor variables was not available. However, the modelling framework addresses uncertainty in the model by calculating the standard deviation of the QRF predictions. Furthermore, the sensitivity of the model to variations in the available data was estimated by means of resampling. To that end, the response data were repeatedly (25 times in this case)
split into training and test subsets at a ratio of 7:3 and 25 models were subsequently built based on these splits. This resampling scheme is known as Monte Carlo cross-validation. The sensitivity is derived by calculating the standard deviation of the 25 predictions for every pixel. The total uncertainty is the sum of the model uncertainty and the sensitivity. The methodology was adapted from Guevara et al. (2018).

Prior to model building, the predictor variables were submitted to a variable selection process. This was achieved via the
Boruta variable selection wrapper algorithm (Kursa and Rudnicki, 2010), which identified important predictor variables. Random forest has been shown to perform well without parameter tuning. Our own experience shows that the gains made by random forest model tuning are comparatively small, while at the same time this step might be time consuming, especially when tuning an array of parameters. As QRF is based on random forest, we assume that the same holds true here. Only limited model tuning was therefore carried out. The number of variables to consider at any given split ($m_{try}$) was tuned in a
grid search using a 10-fold cross-validation scheme with three repeats on the training dataset. It is usually sufficient to set the number of trees in the forest ($n_{tree}$) to a high value; 500 was selected in this case.

The QRF algorithm provides a means of ranking predictor variables by their importance to prediction accuracy. Variable importance is measured as the mean increase in node purity. Node purity represents how well the trees in the forest split the data.

The model performance was assessed based on the test data of 25 resampling iterations. The root mean square error (RMSE) was calculated according to:

$$RMSE = \sqrt{\frac{1}{n} \cdot \sum_{i=1}^{n}(y_i - \hat{y}_i)^2}$$  (3)

RMSE measures how far apart on average predicted values are from observed values. It might range from 0 to infinity, with an ideal value of 0. It is reported in the same units as the predicted quantity. Additionally, the explained variance ($r^2$) was calculated from the observed and predicted values.

The analysis was carried out in R 3.6.1 statistical software (R Core Team, 2018) and RStudio 1.2.1335. The full workflows are documented as R Notebook files (Supplement S1 and S2).

## 4.2 Calculation of OC stocks

The OC stock ($m_{OC}$) of surface sediments in the North Sea and Skagerrak was calculated by summing the predicted OC densities of all pixels and multiplying with the reference depth (d = 0.1 m) and the area of a pixel (A = 250,000 m$^2$):

$$m_{OC} = d \cdot A \cdot \sum \rho_{OC} \tag{6}$$


The total uncertainty of the predicted OC stock was calculated in the same way. OC stocks and uncertainties are reported in Tg C. One Tg C equals 1 Mt C or 0.083 Tmol C.

## 4.3 Calculation of OC accumulation rates

OC accumulation rates (OCAR in g m$^{-2}$ yr$^{-1}$) were calculated by multiplying predicted OC densities with predicted
sedimentation rates:

$$OCAR = \rho_{OC} \cdot \omega \tag{7}$$

Uncertainties were propagated by taking the square root of the sum of squared relative uncertainties:


$$\frac{\delta OCAR}{OCAR} = \sqrt{\left(\frac{\delta \rho_{OC}}{\rho_{OC}}\right)^2 + \left(\frac{\delta \omega}{\omega}\right)^2} \tag{8}$$

Whereby δ denotes the uncertainty of a quantity. The full workflow is documented as an R Notebook file (Supplement S3).

## 4.4 Regionalisation

An unsupervised classification was carried out to provide a regionalisation of the North Sea environment with regard to processing of OC at the seafloor. The following environmental variables were selected: bathymetry, tidal current speed, peak orbital velocity, oxygen penetration depth, OC density and OC accumulation rate. These are expected to have a strong impact on OC processing. A k-means clustering was conducted utilising the algorithm of Hartigan and Wong (1979). Prior to clustering, the input variables were normalised, and a principal component analysis was carried out to limit co-linearity in
the input data. The first four principal components, accounting for 95.5 % of the variance, were selected for further analysis. The selection of the number of clusters to be requested was based on an elbow plot, which resulted in three clusters. The full workflow is documented as an R Notebook file (Supplement S4).

## 5 Results

### 5.1 Sedimentation rates

Of the thirteen predictor variables initially selected for model building (Table 1), only the Folk textural class was found unimportant and hence removed. The five most important predictor variables were the M2 tidal current velocity, the ratio of tidal boundary layer thickness to water depth, the peak orbital velocity, sand content, and mud content (Fig. 3). The selected predictors are shown in Fig. A2.

The model had an RMSE of $0.13 \pm 0.03$ cm $yr^{-1}$, and an $r^2$ of $0.58 \pm 0.09$. Predicted sedimentation rates range from 0 to

0.61 cm $yr^{-1}$, while the total uncertainty varies between 0.12 and 0.53 cm $yr^{-1}$ (Fig. 4). Sedimentation rates are highest in the Norwegian Trough. Zero net sedimentation occurs in large parts of the North Sea, with slightly elevated sedimentation rates linked to shallow basins such as the inner German Bight, the Elbe palaeo-valley, the Oyster Ground, the Outer Silver Pit and Devil's Hole. The patterns of prediction uncertainty follow those of the sedimentation rate.

### 5.2 Organic carbon density

All thirteen predictor variables initially selected for model building (Table 1) were deemed important. The five most important predictor variables were bathymetry, sedimentation rate, bottom water temperature, oxygen exposure time, and mud content (Fig. 3). The selected predictors are shown in Fig. A3.

The model had an RMSE of $2.16 \pm 0.25$ kg $m^{-3}$, and an $r^2$ of $0.72 \pm 0.06$. Predicted OC densities range from 1.11 to 13.59 kg $m^{-3}$, while the total uncertainty varies between 0.89 and 8.07 kg $m^{-3}$ (Fig. 5). OC densities are highest in the

Norwegian Trough. Intermediate OC densities are found in the northern North Sea and shallow basins, while they are lowest on Dogger Bank, in the Southern Bight and along the Danish coast. Note that uncertainties in parts of the Norwegian Trough are comparatively low due to a high sampling density (Figure 2).

The OC stock of surface sediments of the North Sea and Skagerrak amounts to $230.5 \pm 134.5$ Tg C, of which $60.1 \pm 18.3$ Tg C are stored in the Norwegian Trough. This means that 25.9 % of the total OC stock is located within the Norwegian

Trough, which accounts for 11 % of the surface area.

### 5.3 Organic carbon accumulation rates

OC accumulation rates vary between 0.02 and 66.18 g $m^{-2}$ $yr^{-1}$, while the total uncertainty ranges from 0.20 to 57.90 g $m^{-2}$ $yr^{-1}$ (Figure 6). OC accumulation rates are effectively zero over large parts of the North Sea. Marked accumulation of OC is restricted to the Norwegian Trough, which accumulates $1.24 \pm 1.30$ Tg C $yr^{-1}$. This accounts for nearly 87% of the total OC

accumulation of $1.43 \pm 2.07$ Tg C $yr^{-1}$ in the North Sea and Skagerrak.

### 5.4 Regionalisation

The unsupervised classification resulted in regions that were distinct regarding bathymetry, hydrodynamics, oxygen penetration and OC (Fig. 7). Region 2 (green) is characterised by shallow water, strong hydrodynamics, deep oxygen penetration, low OC densities and OC accumulation close to zero. Region 3 (dark blue) is characterised by deep water, weak

hydrodynamics, shallow oxygen penetration, high OC densities and high OC accumulation. Region 1 (light blue) has characteristics that lie intermediate between those of regions 2 and 3.

## 6 Discussion

We have presented estimates of OC stocks and accumulation rates and their associated spatially explicit uncertainties that were derived with the same modelling framework. Our results show that a substantial amount of OC, 231 Tg C within the
upper 0.1 m of seabed sediment, is stored in surface sediments of the North Sea and Skagerrak. OC accumulation is effectively restricted to the Norwegian Trough, which accumulates 1.2 Tg C annually. In the following we discuss the relevance of our results by comparing them with other estimates of OC stored in shelf sea sediments, coastal vegetated habitats, and terrestrial soils, which have been highlighted as significant OC stores. We further discuss zones of OC processing at the seafloor based on our regionalisation, potential implications for marine management and suggestions for
future research.

### 6.1 Relevance

The surface sediments of the North Sea and Skagerrak store 230.5 ± 134.5 Tg of OC. This compares with 9.6 to 25.0 Pg C stored globally in bioturbated Holocene shelf sediments (0 – 10 cm) as estimated by LaRowe et al. (2020). Hence, sediments in the North Sea and Skagerrak store approximately 0.9 – 2.4 % of the global stock in an area that accounts for ≈ 1.7 % of
the global shelf.

When comparing uncertainties in OC stock estimates with other reported values of spatial predictions at a regional to global scale, we find that our value of 58 % (100 * 134.5 Tg  C/ 230.5 Tg C) is similar to that reported by Lee et al. (2019) amounting to 49 %, while other studies did not report any estimates of uncertainty (Diesing et al., 2017; LaRowe et al., 2020). Lower uncertainties have been reported from local studies (e.g. Hunt et al., 2020), presumably due to a tighter
coupling between response and predictor variable. An intrinsic assumption of modelling approaches such as the one presented here is that the measured response variable is representative at the scale of the pixel size of predictor variables. The likelihood for this being true increases when the pixel size approaches the size of the seabed area that was sampled with a grab or corer. Higher resolution predictor variables, as frequently used in local studies, might therefore have lower uncertainties associated with the predictions. It should also be considered that the ways in which uncertainty is estimated and
reported vary, thereby limiting the scope of such comparisons. We believe that our approach to uncertainty assessment is very robust as it estimates uncertainty in the model and in variations of available data.

Previous estimates of OC stocks in the upper 10 cm of the sediment column of the northwest European continental shelf amount to 230 – 882 Tg C (Diesing et al., 2017). The estimated stock of 230.5 ± 134.5 Tg C contained in the upper 10 cm of the sediments of the North Sea and Skagerrak, which account for approximately 50 % of the area of the North-West

European continental shelf, falls well within this estimate. Of this stock approximately 60 Tg C or 26 % are stored within the Norwegian Trough, indicating the importance of this glacial feature as a store of OC.

To gauge the importance of North Sea shelf sediments as an OC store, we compare them with coastal habitats and terrestrial soils as follows: Coastal vegetated habitats (saltmarsh, seagrass, kelp and tidal flat) are known to bury large amounts of carbon despite occupying only 0.2 % of the global ocean surface (Duarte et al., 2005, 2013). Coastal habitats on the northwest European continental shelf store between 8.3 and 40.8 Tg C in the upper 10 cm in an area of 20,900 – 35,000 $km^2$ (Legge et al., 2020), equating to OC densities between 24 and 195 $kg\,m^{-3}$. This indicates that shelf sediment stocks (230.5 Tg C) are approximately an order of magnitude larger despite lower OC densities of 1.1 to 13.6 $kg\,m^{-3}$

Soils are the largest carbon store on land; globally they are estimated to hold 1325 Pg C in the upper 1 m (Köchy et al., 2015). Topsoil (0 – 10 cm) OC stocks based on SoilGrids250m (Hengl et al., 2017) of the countries bordering on the North Sea and Skagerrak are shown in Table 2. Note that topsoil OC stocks refer to the entire area of the respective country, while marine OC stocks refer to the proportion of the EEZ that falls within our study area. While marine sediment OC stocks are generally lower than their soil counterparts, marine stocks are not negligible in several countries. These additional OC stocks amount to 7.5 %, 7.1 % and 6.6 % of topsoil stocks in Denmark, the Netherlands, and the United Kingdom, respectively. Furthermore, some countries have Exclusive Economic Zones (EEZs) considerably larger than their share of the North Sea and Skagerrak considered here. Hence, there is potential for an even larger marine OC stock.

The accumulation of OC is effectively limited to the Norwegian Trough, with the highest rates found in the Skagerrak. Predicted OCARs vary between approximately 4 and 66 $g\,m^{-2}\,yr^{-1}$ in the Norwegian Trough, with a mean OCAR of 19.4 $g\,m^{-2}\,yr^{-1}$. Reported OCARs measured in fjord sediments in Norway and Sweden bordering on the North Sea range from 12 to 54 $g\,m^{-2}\,yr^{-1}$ (Huguet et al., 2007; Müller, 2001; Nordberg et al., 2001, 2009; Skei, 1983; Smittenberg et al., 2004, 2005; Velinsky and Fogel, 1999), indicating that OCARs in the Norwegian Trough are of a comparable magnitude. However, fjords in Scotland and Ireland have been shown to be heterogeneous in sediment distribution and OC concentrations (Smeaton and Austin, 2019), and hence also OC accumulation. Judging from published sediment maps (e.g. Elvenes et al., 2019), the same applies to fjords in Norway. Conversely, the Norwegian Trough is characterised by fine-grained sediments (Mitchell et al., 2019) and OC accumulation occurs throughout the geomorphological structure. Additionally, the area of the Norwegian Trough is much larger than even the largest fjords in Norway, highlighting its relevance as the most important place of OC accumulation in the North Sea and Skagerrak.

Collectively, the sediments of the Norwegian Trough accumulate 1.24 Tg C $yr^{-1}$ over an area of approximately 62,000 $km^2$, but the uncertainty in this estimate is on the same order of magnitude as the estimate. Nevertheless, this estimate is in good agreement with an earlier published value of 1 Tg C $yr^{-1}$ (de Haas and van Weering, 1997). For comparison, 3.53 ± 2.90 Tg C $yr^{-1}$ are accumulated in the muddy basins of the Baltic Sea (area: 164,800 $km^2$) (Leipe et al., 2011). Coastal habitats (saltmarsh, seagrass, kelp and tidal flat) on the northwest European continental shelf have been estimated to accumulate 0.2 – 0.7 Tg C $yr^{-1}$ (Legge et al., 2020).

## 6.2 Zones of organic carbon processing at the seafloor

The regionalisation based on selected characteristic parameters pertaining to OC accumulation and storage (Fig. 7) has shown that the North Sea and Skagerrak can be divided into distinct zones. The results indicate that shelf sediments can act in distinctly different ways in the context of OC processing at the seafloor. In a way, they also reflect the scientific discourse over the last half century or so: Initially, process studies on OC cycling on the continental shelf (e.g. Balzer, 1984; Jørgensen, 1977; Martens and Val Klump, 1984) focussed on fine-grained sediments associated with hydrodynamically quiet environments, relatively constant sediment accumulation and diffusion-dominated porewater transport. This has led to the notion of rapidly accumulating coastal sediments associated with high sedimentation, high OC burial rates and low oxygen penetration depths (Aller, 2014; Canfield, 1994; Middelburg, 2019; Middelburg et al., 1997).

However, approximately 50 % (Hall, 2002) to 70 % (Emery, 1968) of the global shelf consist of coarse-grained sediments (gravel and sand) with high permeabilities. Unidirectional and wave orbital water flows interacting with microscale topography (e.g. ripples and biogenic mounds) at the water-sediment interface lead to increased fluid exchange rates compared to exchange by molecular diffusion (Huettel et al., 1996; Precht and Huettel, 2003). Interaction of flows with surface microtopography increases oxygen penetration depths (Huettel and Rusch, 2000). As a consequence of advective porewater flows, permeable sediments may act as biocatalytic filters, notable for their high reaction rates, intense recycling, and extreme spatial and temporal dynamics of biogeochemical processes (Huettel et al., 2003, 2014).

The seafloor in the Southern Bight, on Dogger and Fisher Banks and in the proximity to west-facing coastlines (apart from the Norwegian west coast) is characterised by shallow water depths, high tidal current speeds, and high wave orbital velocities. The probability that the seabed gets disturbed by waves and currents to a depth of 3 cm at least once a year is above 50 % in these areas (Aldridge et al., 2015: Fig. 17a). It can therefore be assumed that ripples are present in these areas at least temporarily and that the interaction of unidirectional and oscillatory currents with these roughness elements leads to enhanced fluid exchange, as sediments are sufficiently permeable. The advective supply of oxygen to the sedimentary microbial community facilitates the effective degradation of OC (Huettel et al., 2014). Consequently, oxygen penetrates deep into these sediments and OC density is low. The potential for longer-term accumulation of OC is very low, as these environments are characterised by repeated erosion-redeposition cycles. This zone of rapid OC processing might equate to the turnover zone of Huettel and Rusch (2000).

Conversely, the seabed of the Norwegian Trough is characterised by water depths in excess of 200 m and experiences very subdued wave and current agitation. Fluid transport in the sediment is therefore driven by molecular diffusion, mediated by bioturbation. Bioturbation contributes to a balance in the sedimentary OC budget by transporting labile OC to deeper horizons where degradation efficiency is lower (Zhang et al., 2019). The lack of advective oxidation (Huettel et al., 2014; Huettel and Rusch, 2000) translates into slower OC degradation. Fine-grained sediments provide mineral protection (Hedges and Keil, 1995; Hemingway et al., 2019; Keil and Hedges, 1993; Mayer, 1994), which also promotes OC preservation. Short oxygen exposure times (Hartnett et al., 1998) due to shallow oxygen penetration depths and relatively high sedimentation

rates limit the time for aerobic mineralisation. Collectively, this leads to high OC densities and accumulation rates. This zone might be termed a burial zone according to Huettel and Rusch (2000). However, for consistency with our analysis we term this zone an accumulation zone.

De Haas and van Weering (1997) estimated that only 10 % of the OC deposited in the Norwegian Trough is derived from local primary production and the remainder originates from other sources. A large part of this allochthonous OC is transported into the Norwegian Trough along the Dutch, German and Danish coasts by an anti-clockwise residual circulation (de Haas et al., 2002). This transport is thought to be intermittent, with the rate of transport dependent on the strength of wind-induced waves and currents (de Haas and van Weering, 1997). The OC being deposited in the Norwegian Trough is mostly refractory, as it has undergone several erosion-transport-deposition cycles prior to final deposition (de Haas et al., 2002).

A third zone is situated in the northern North Sea and the shallow depositional areas of the southern North Sea. It has a transitional character with water depths, current speeds, wave orbital velocities and oxygen penetration depths intermediate between those of the turnover and the burial zones. OC densities are also intermediate, while OC accumulation is negligible in this transitional zone.

## 6.3 Implications for management

We have shown that seabed sediments of the North Sea and Skagerrak are an important store of OC. Furthermore, the Norwegian Trough is an important centre of OC accumulation, with rates comparable with neighbouring fjords. Based on those results it was possible to identify zones of rapid OC turnover and zones of OC accumulation. These zones have different roles in terms of OC processing and storage and hence will have different relevance in the context of managing OC stores at the seabed.

Marine sediment OC stocks are presently not considered in the context of national carbon inventories for greenhouse gas reporting. The question has been raised whether those stocks should be considered as part of national carbon accounting (Avelar et al., 2017). It is becoming clearer that marine sediments store sizeable amounts of OC (Diesing et al., 2017; Lee et al., 2019; Luisetti et al., 2019), which might be vulnerable to human activities such as demersal fishing (Paradis et al., 2020). Likewise, there exist hot spots of OC accumulation (Bianchi et al., 2018) like the Norwegian Trough, as demonstrated here. A further exploration as to how management of marine sediment OC could contribute towards national greenhouse gas emission reduction targets might therefore be prudent; however, this requires new accounting guidance and governance frameworks (Luisetti et al., 2020). The assessment of the OC stock size should be coupled with an assessment of the anthropogenic impacts on that stock (Avelar et al., 2017). When assessed in the context of naturally occurring disturbance (e.g., by currents and waves), this will contribute towards a more complete picture of the vulnerability of marine sediment OC stocks to remineralisation and potential release of $CO_2$ to the atmosphere (Atwood et al., 2020). We provide spatially explicit information on stock sizes and the uncertainty in the estimates, which could be utilised in such vulnerability assessments.

While the importance of Blue Carbon ecosystems for OC drawdown has been highlighted in the past (Duarte et al., 2005;
Mcleod et al., 2011; Nellemann et al., 2009), the annual rate of OC accumulation by coastal vegetated habitats (Legge et al., 2020)is less than that of seafloor sediments at a sea basin scale. It might therefore be prudent to further explore the idea of MPAs as a tool to mitigate climate change by protecting and enhancing marine sedimentary OC stores (Roberts et al., 2017), especially as the climate mitigation potential of marine natural climate solutions (Griscom et al., 2017) has so far been overlooked.

Although more research is needed, it is becoming clearer now that seabed disturbance by demersal fishing leads to increased OC mineralisation in cohesive sediments in the short-term (van de Velde et al., 2018) and a general impoverishment in OC in the long-term (Martín et al., 2014a). Protecting regional hotspots of OC accumulation from fishing-induced disturbance might therefore be a suitable measure to increase the climate mitigation potential of the seabed. Likely sites that might benefit from protection are to be found in the accumulation zone (i.e., the Norwegian Trough), while it is unlikely that the

turnover zone yields any potential areas worth protecting in this context. Our results could be used jointly with maps showing the footprint of demersal fishing (Eigaard et al., 2016) and other resources to identify potential sites for the establishment of "carbon protection zones". Such management measures that limit the impacted surface area, allowing carbon stocks and faunal communities in the sediment to recover from a disturbance, and resulting in the recovery of carbon burial, might be preferable over technical modifications that reduce the penetration depth of fishing gear (De Borger et al.,

2020). Recent research also highlights that temporal closures of fishing grounds might not be sufficient to restore the seafloor (Paradis et al., 2020). It must also be considered that the OC stocks, as mapped in this study, likely have been affected already by decades of demersal fishing. Our maps therefore do not represent a baseline in a sense of an undisturbed state.

Additionally, more research on the reactivity of OC is required to better understand the relationships between OC
mineralisation and seabed disturbance. The mineralisation of predominantly refractory OC caused by demersal fishing might be limited or even negligible. In the Skagerrak, oxygen microprofile measurements indicated that mineralisation rates were independent of OC content, but related to the input of fresh OC by primary production (Bakker and Helder, 1993). This suggests that preferentially fresh labile OC was mineralised, while allochthonous OC that accounts for 90% of the OC in the Norwegian Trough (de Haas and van Weering, 1997) might be largely unreactive. Conversely, van de Velde et al. (2018)

suggested that OC mineralisation is stimulated after sediment disturbance, likely due to the enhanced decomposition of previously buried refractory OC when it comes into contact with labile OC, a process known as priming (Steen et al., 2016). Another question of interest is to what extent a potential reduction in mineralisation rates due to areal protection of OC stocks might influence primary production and thus supply of OC to the seabed.

**6.4 Suggestions for future research**

We have utilised a modelling scheme that allowed us to estimate the uncertainty in the model and in variations of available data. However, this robust methodology led to relatively high uncertainties in the predictions. We assume that the most

likely reason for this is the nature of the available sample datasets. As we utilised archived samples collected over many years by different organisations for various purposes, this has led to a somewhat heterogeneous dataset with biases regarding coverage of the temporal, geographical and environmental (i.e., predictor variable) space. While we believe that making best use of existing data is important and yields worthwhile insights, this study also highlights the limitations of such an approach. Consequently, there is a need for the collection and analysis of new samples on OC content, dry bulk density, sedimentation rates and ancillary parameters (e.g., grain size). Sampling design might be guided by the uncertainty maps provided here. The information gain additional data could give is expected to be highest in areas of high predictive uncertainty. Reducing uncertainty in predictions might have large economic benefits, as has recently been demonstrated for the biological carbon pump (Jin et al., 2020). These authors developed an analytical model of the economic effects of global carbon emissions including uncertainty about biological carbon pump sequestration and estimated that the benefit to narrow the range of uncertainty about ocean carbon sequestration is on the order of $ 0.5 trillion. It may be assumed that sizeable economic benefits could also be achieved by reducing the uncertainty in the predictions of seafloor OC stocks and accumulation rates.

Alternatively, if the goal were to create a new baseline dataset covering the whole North Sea and Skagerrak, this might be best achieved by sampling the environmental variable space in a representative way. The relative importance of environmental variables on the distribution of OC is relatively well known both based on general knowledge and the results of this and other modelling studies. Several methods for optimising sampling design exist, including generalised random tessellation stratified (Stevens Jr and Olsen, 2003) and conditioned Latin hypercube sampling (Minasny and McBratney, 2006) among others. These could be utilised to effectively sample seafloor sediments, thereby minimising sampling effort and prediction uncertainty at the same time. Finally, future process studies might compare results from different zones of OC processing as shown in Figure 7.

Further gains could be achieved by the standardisation of the collection of OC measurements. This includes sampling methods, the measured sediment fraction, defined depth horizons, and the reporting of results, among others. Such a standardisation would increase the comparability of the collected data and could be modelled on the experience of the global soil mapping community (Hengl et al., 2014). Although facilities to store and retrieve quality-controlled seafloor data centrally exist (e.g., EMODnet, ICES), it would still be advantageous to establish global data archives that are more specific to marine sedimentary carbon such as MOSAIC (van der Voort et al., 2020).

Finally, it would be desirable to complement OC data with measurements on C/N-ratios and $\delta^{13}C$ to estimate the marine versus terrigenous fraction of OC (e.g. Faust and Knies, 2019). A quantification of the autochthonous and allochthonous OC contributions could be achieved with a two-end-member mixing model (Thornton and McManus, 1994). Knowledge of the sources of OC is required for a better understanding of OC sequestration in shelf sediments, but would also be a basic requirement in the context of carbon offset-credits (Macreadie et al., 2019), should such a system be extended to include shelf sediments. For example, the Verified Carbon Standard VM0033 (https://verra.org/methodology/vm0033-methodology-for-tidal-wetland-and-seagrass-restoration-v1-0/), the first voluntary market methodology for blue carbon ecosystems,

stipulates that offset-credits are not allocated under the framework for allochthonous OC because of the risk of duplicating C sequestration gains that may have been accounted for in adjacent ecosystems (Macreadie et al., 2019).

**7 Conclusions**

485   This work highlights distinct zones of OC processing at the seafloor of the North Sea and Skagerrak. While rapid OC processing and turnover are commonplace in the southern and eastern parts of the North Sea, the Norwegian Trough stands out as a hotspot of OC accumulation with rates comparable with nearby fjords. We expect that this dual character of the continental shelf in terms of OC processing and storage can be found across the global shelf, requiring further detailed and spatially explicit analyses to constrain sedimentary OC stocks and accumulation rates globally. Such estimates are urgently needed to better understand the potential of shelf sediments as a natural climate solution, e.g. by protecting suitable areas

490   against human disturbance.

**Code availability**

The following R Markdown documents are provided as supplements:

Supplement S1: Sedimentation Rates and Uncertainty

Supplement S2: Organic Carbon Density and Uncertainty

495   Supplement S3: Organic Carbon Accumulation Rates and Uncertainty

Supplement S4: Regionalisation of the Environment

**Data availability**

Grids of sedimentation rates, OC density, OC accumulation rates and their associated uncertainties are available on the PANGAEA website: https://doi.org/10.1594/PANGAEA.928272.

500   **Sample availability**

The following data tables are provided as supplements:

Supplementary Data Table 1: Linear sedimentation rates

Supplementary Data Table 2: Organic carbon densities

**Author contribution**

Conceptualization – MD, TT, LRB; Data curation – MD; Formal Analysis – MD; Funding acquisition – TT, LRB; Investigation – MD; Methodology – MD; Project administration – LRB, TT; Validation – MD; Visualization – MD; Writing – original draft – MD; Writing – review & editing, MD, TT, LRB

**Competing interests**

The authors declare that they have no conflict of interest.

**Acknowledgements**

This study was financed by the Norwegian seabed mapping programme MAREANO.

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

 **Table 1: Predictor variables used in the sedimentation rate and OC density models.**

| Predictor variable (Abbreviation) | Unit | Model | Source |
|---|---|---|---|
| Bathymetry (Bathy) | m | Both | EMODnet Bathymetry Consortium (2018) Mitchell et al. (2019) |
| Euclidean distance to shoreline (DistCoast) | m | Both | Calculated |
| Mud content (Mud) | % | Both | https://doi.org/10.14466/CefasDataHub.63 Mitchell et al. (2019) |
| Sand content (Sand) | % | Sedimentation rate | https://doi.org/10.14466/CefasDataHub.63 Mitchell et al. (2019) |
| Gravel content (Gravel) | % | Sedimentation rate | https://doi.org/10.14466/CefasDataHub.63 Mitchell et al. (2019) |
| Folk textural class (Folk) | - | Sedimentation rate | https://doi.org/10.14466/CefasDataHub.63 Mitchell et al. (2019) |
| Summer suspended particulate matter (SPM_summer) | g m$^{-3}$ | Both | http://marine.copernicus.eu/ Mitchell et al. (2019) |
| Winter suspended particulate matter (SPM_winter) | g m$^{-3}$ | Both | http://marine.copernicus.eu/ Mitchell et al. (2019) |
| M2 tidal current speed (M2Speed) | m s$^{-1}$ | Both | https://doi.org/10.14466/CefasDataHub.62 Mitchell et al. (2019) |
| Peak orbital velocity (PkOrbVel) | m s$^{-1}$ | Both | https://doi.org/10.14466/CefasDataHub.62 Mitchell et al. (2019) |
| Ratio of tidal boundary layer thickness to water depth (delta_star) | - | Sedimentation rate | Williams et al. (2019) |
| Geomorphology (Geomorph) | - | Sedimentation rate | Derived from Harris et al. (2014) |
| Sedimentary environment (SedEnv) | - | Sedimentation rate | Derived from Mitchell et al. (2019) |
| Mean bottom water oxygen (O2_mean) | mol m$^{-3}$ | OC density | http://www.bio-oracle.org/index.php Assis et al. (2018), Tyberghein et al. (2012) |
| Oxygen penetration depth (OPD)$\delta$ | cm | OC density | Calculated from mud content (pers. comm. John Barry, Cefas): |

| | | | |
|---|---|---|---|
| | | | $\delta = e^{(1.0745 - 0.1431 \cdot mud)}$ for mud $\leq 8.0$ %<br><br>$\delta = e^{-0.0706}$ for mud $> 8.0$ % |
| *Oxygen exposure time (OET)* | yr | OC density | Calculated from sedimentation rate and oxygen penetration depth:<br><br>$OET = \dfrac{\delta}{\omega}$ |
| *Mean bottom water temperature (Temp_mean)* | ºC | OC density | http://www.bio-oracle.org/index.php<br>Assis et al. (2018), Tyberghein et al. (2012) |
| *Mean sea surface primary production (SurfPP_mean)* | g m$^{-3}$ day$^{-1}$ | OC density | http://www.bio-oracle.org/index.php<br>Assis et al. (2018), Tyberghein et al. (2012) |
| *Sedimentation rate (SedRate)* | cm yr$^{-1}$ | OC density | Modelled (this study) |

**Table 2: Breakdown of topsoil (0-10 cm) OC stocks by country (Hengl et al., 2017), compared with marine sediment OC stocks. Topsoil OC stocks refer to the entire area of the respective country bordering on the North Sea and Skagerrak, while marine OC stocks refer to the proportion of the EEZ that falls within our study area.**

| Country | Soil OC (0-10 cm), Tg C | Marine sediment OC (0-10 cm), Tg C | Marine sediment OC % of Soil OC | Mapped area % of total EEZ |
|---|---|---|---|---|
| **Belgium** | 109.3 | 0.7 | 0.7 | 95.3 |
| **Denmark** | 236.6 | 17.8 | 7.5 | 55.1 |
| **France** | 2026.1 | 0.4 | 0.0 | 0.5 |
| **Germany** | 1808.9 | 8.8 | 0.5 | 64.4 |
| **Netherlands** | 198.0 | 14.1 | 7.1 | 91.5 |
| **Norway** | 2253.6 | 83.9 | 3.7 | 13.7 |
| **Sweden** | 3333.2 | 5.0 | 0.1 | 3.7 |
| **United Kingdom** | 1572.3 | 103.1 | 6.6 | 32.7 |

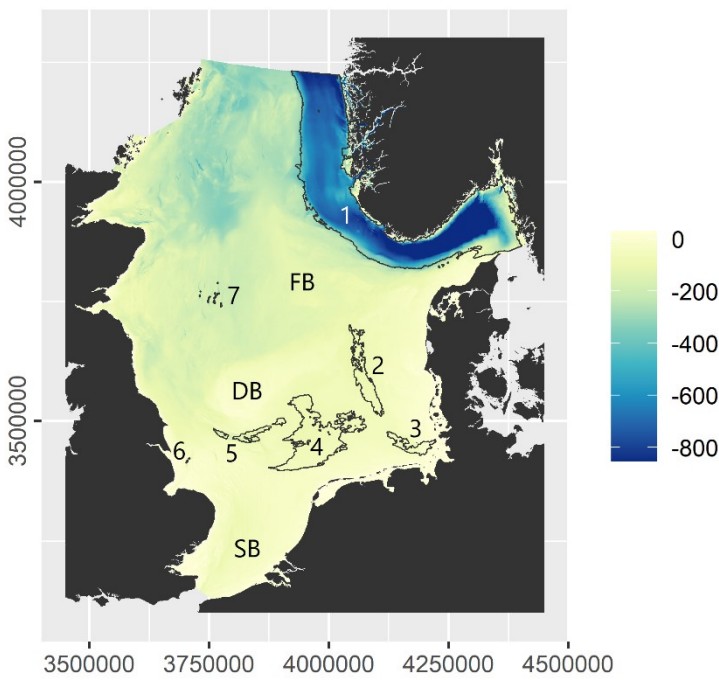

**Figure 1: Overview of the study site. Letters refer to localities: DB – Dogger Bank; FB – Fisher Bank; SB – Southern Bight. Numbers refer to areas of sediment deposition: 1 – Norwegian Trough; 2 – Elbe palaeo-valley; 3 – German Bight; 4 – Oyster Ground; 5 – Outer Silver Pit; 6 – Inner Silver Pit; 7 – Devil's Hole. Refer to chapter 3.2 for the delineation of areas of sediment deposition.**


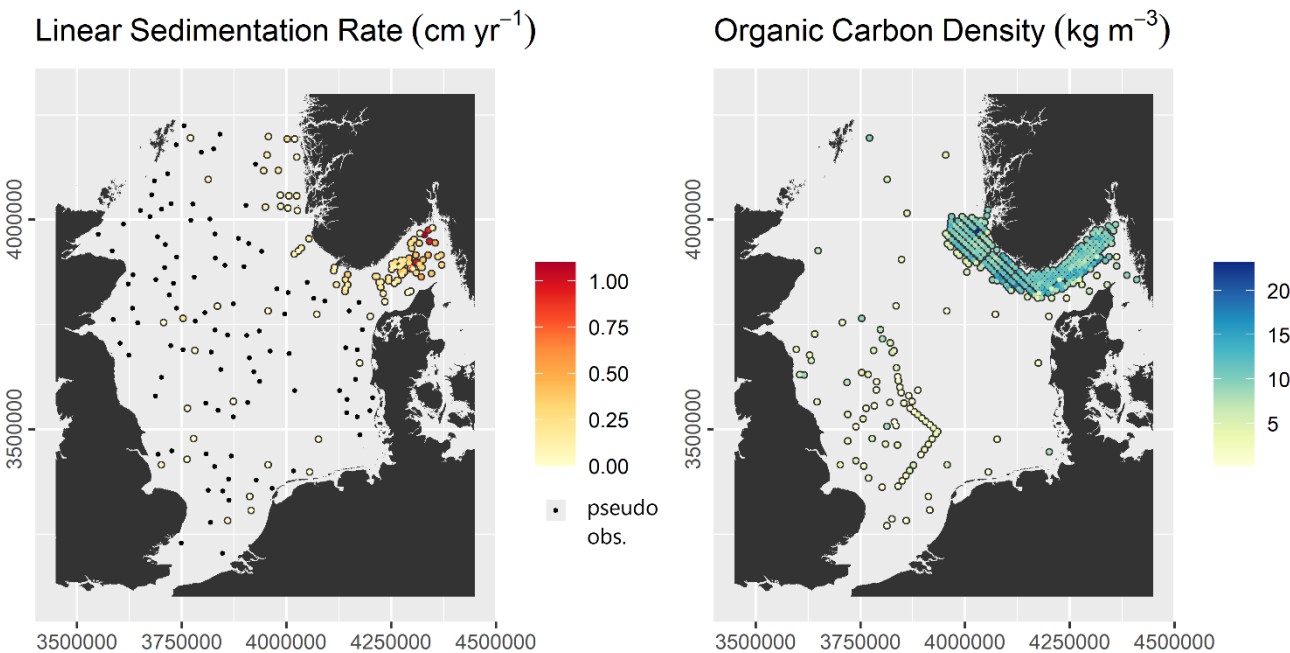

**Figure 2: Available samples on sedimentation rate (left) and OC density (right)**

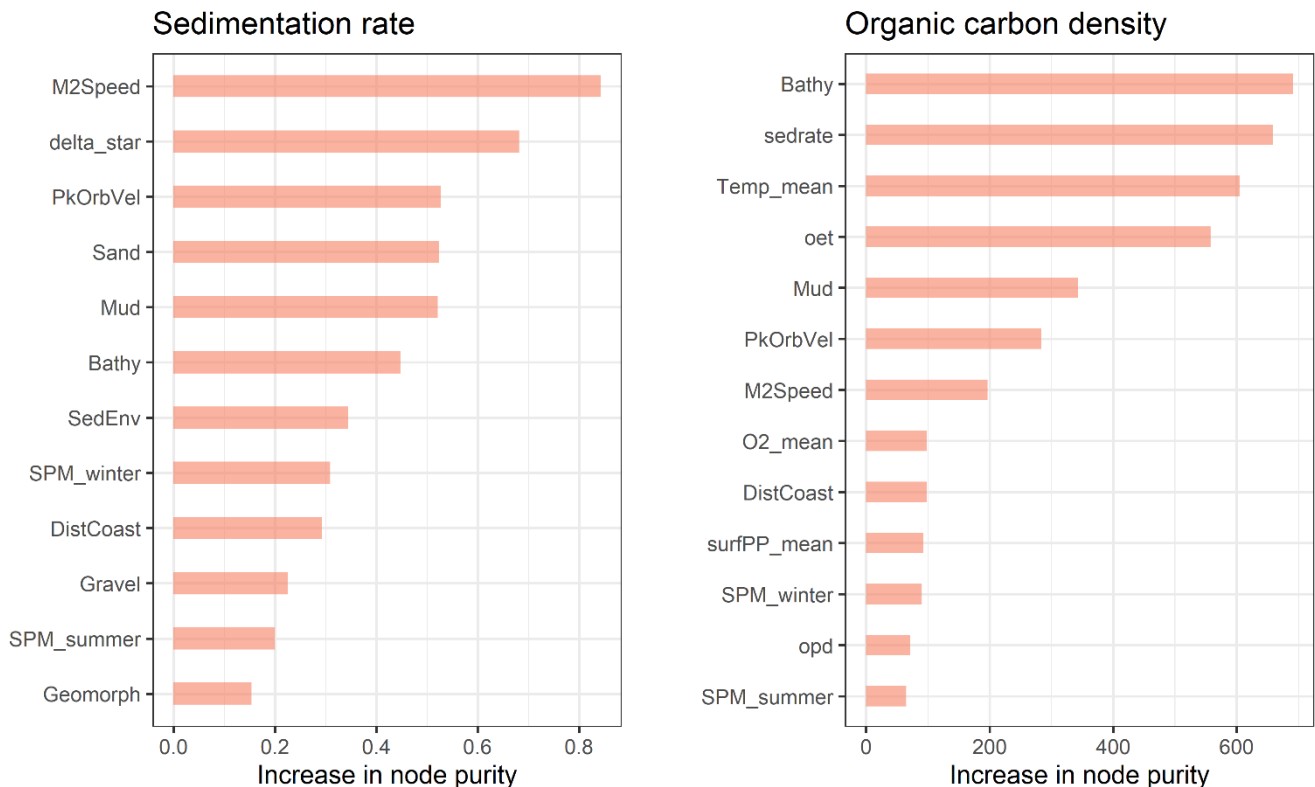

**Figure 3: Selected predictor variables and relative variable importance of the sedimentation rate (left) and organic carbon density (right) models.**


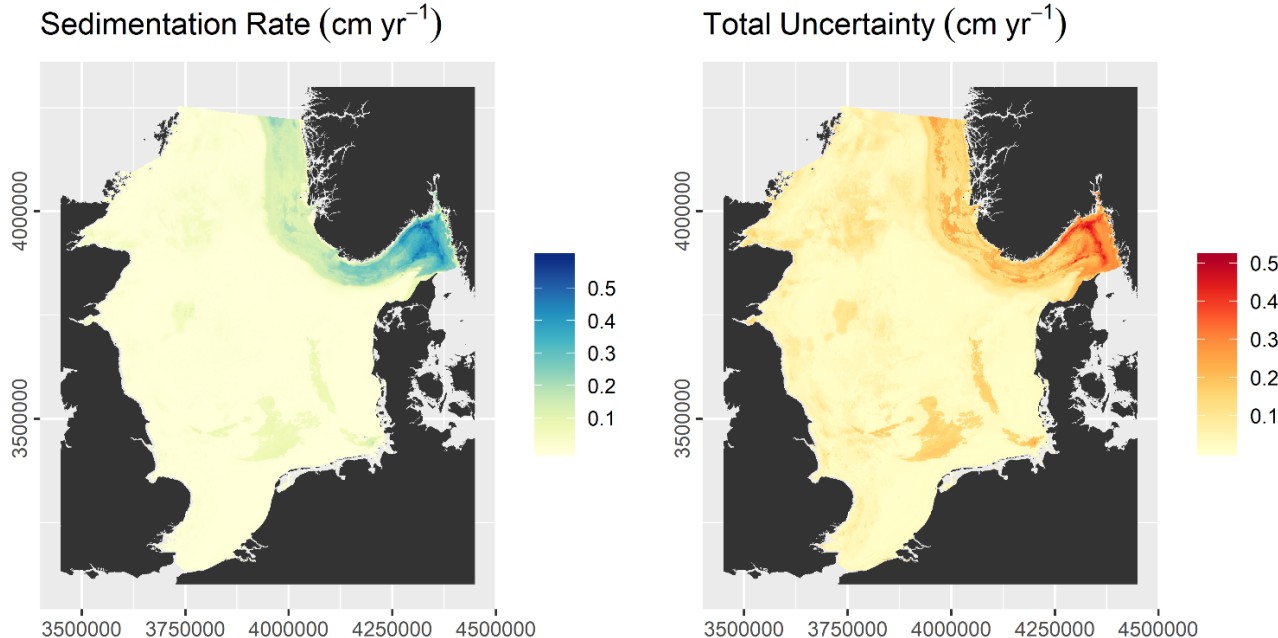

Figure 4: Predicted sedimentation rate (left) and associated uncertainty in the predictions.

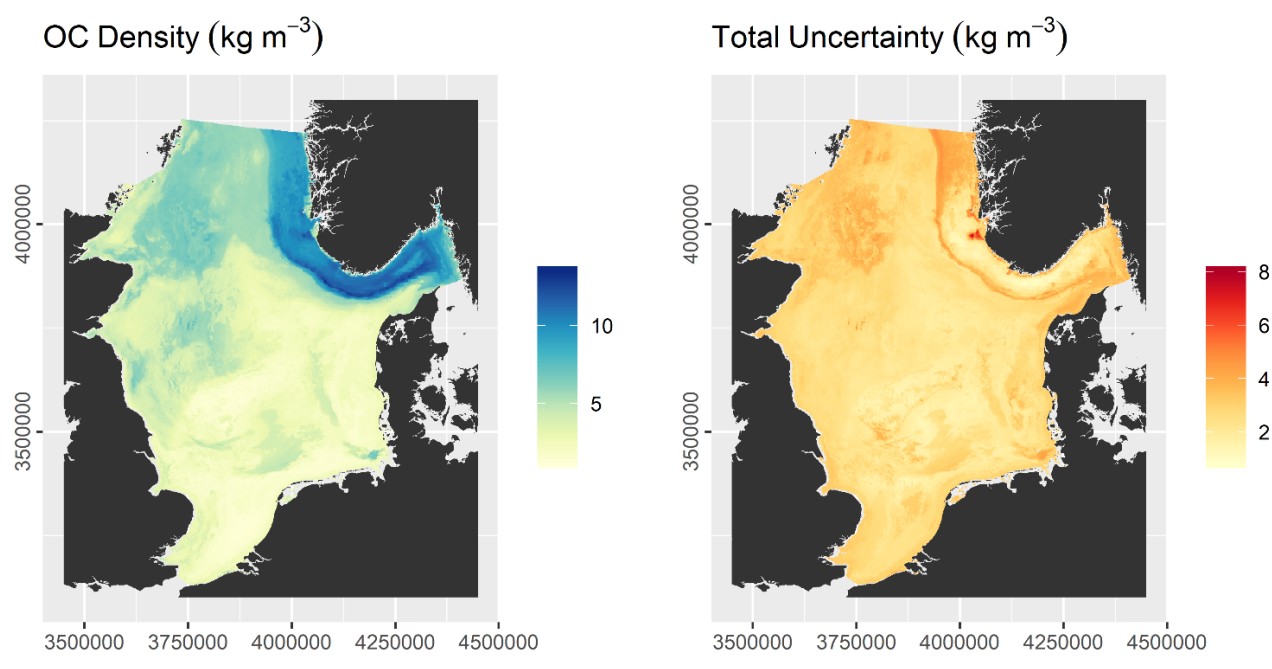


Figure 5: Predicted OC density (left) and associated uncertainty in the predictions (right).

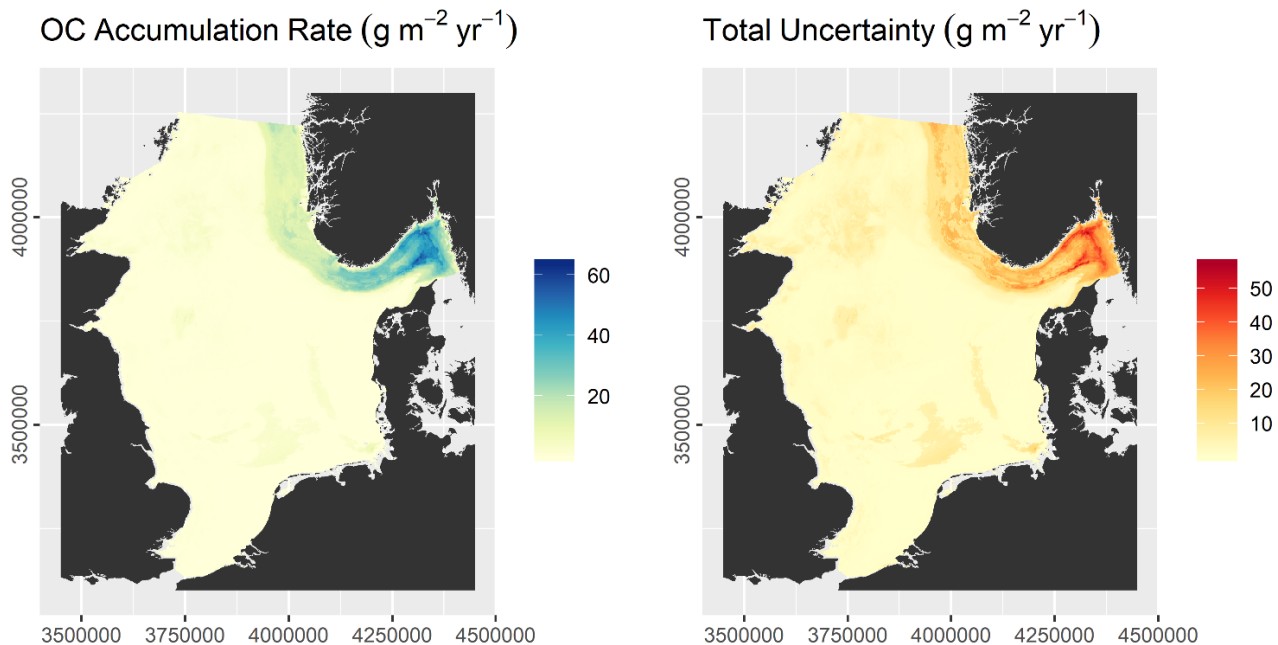

**Figure 6: Calculated OC accumulation rate (left) and associated uncertainty (right)**


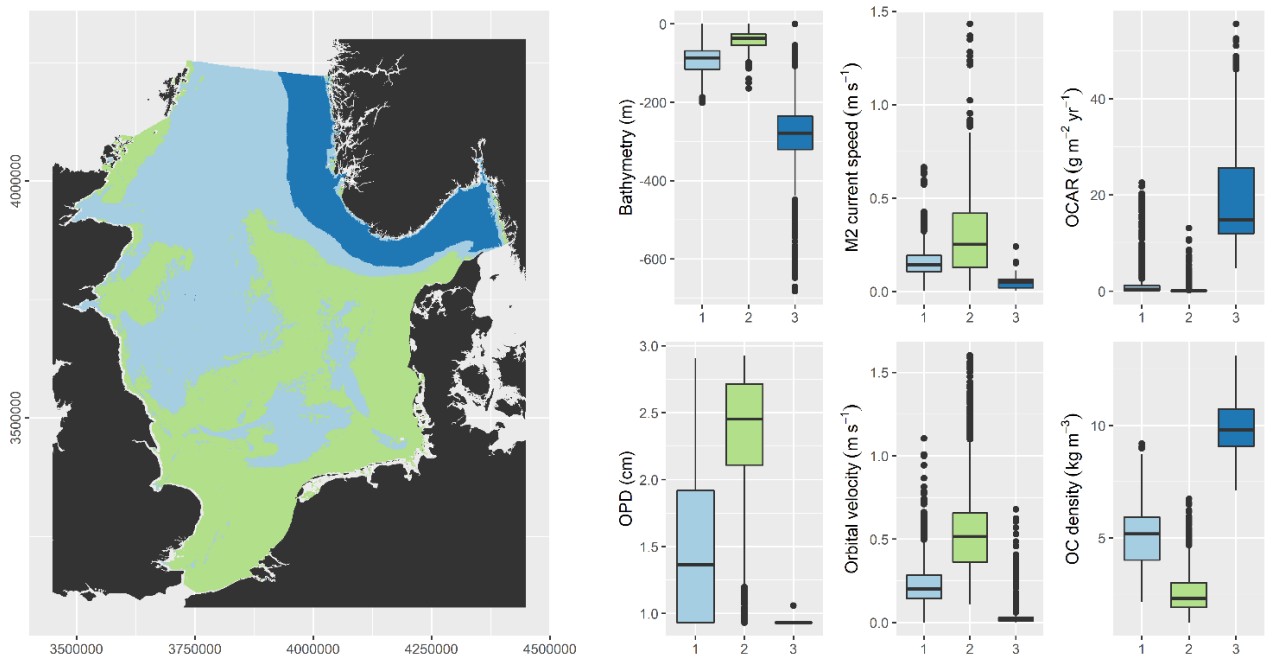

**Figure 7: Regionalisation of the North Sea and Skagerrak: Region 1 (light blue) – transition zone; Region 2 (green) – turnover zone; Region 3 (dark blue) – burial zone. Note that boxplots are based on 10000 randomly placed points, rather than all pixels.**

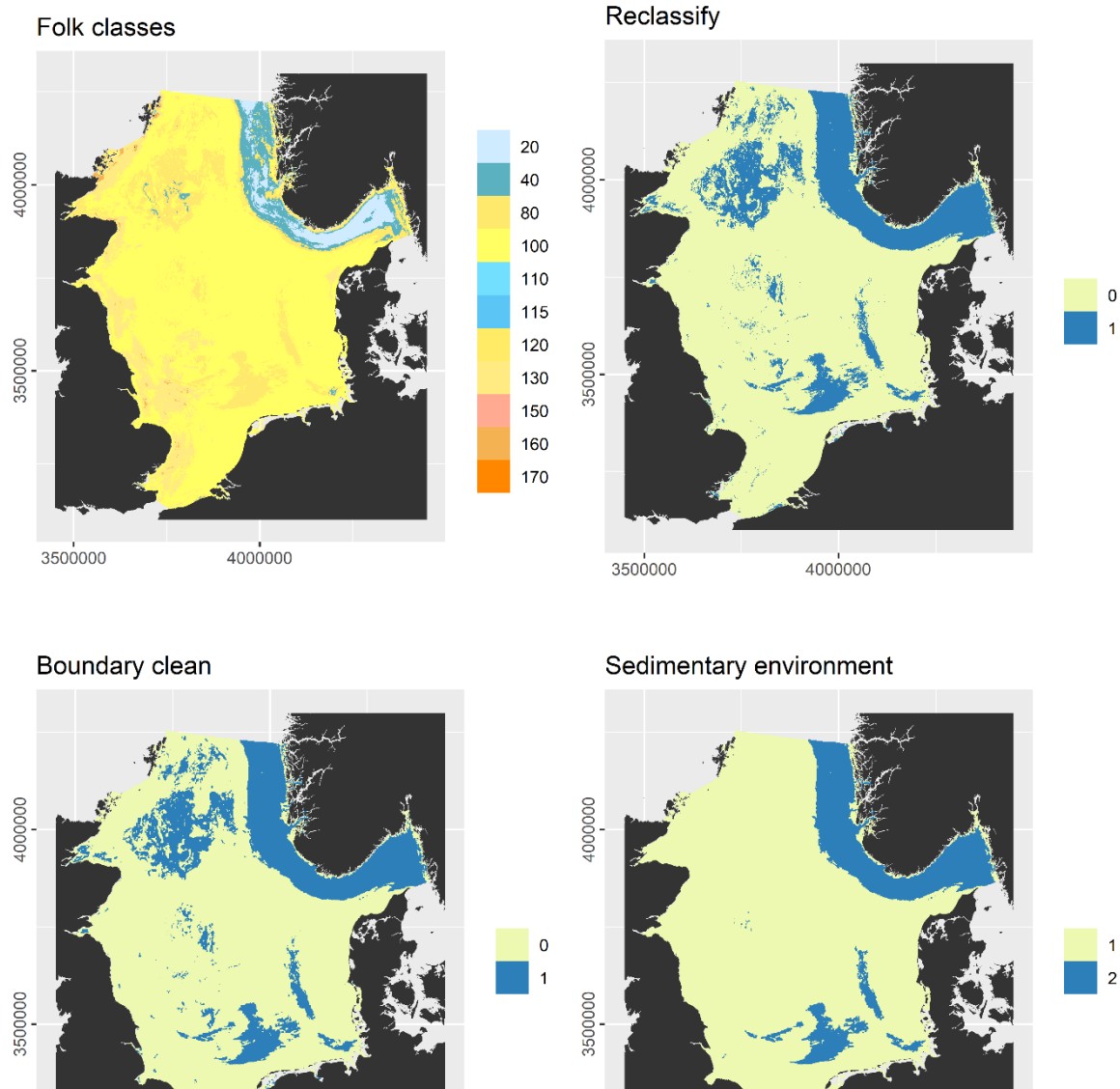

Figure A1: Derivation of the sedimentary environment layer: Folk textural classes were derived from sediment composition predicted by Mitchell et al. (2019). 20 – Mud, 40 – Sandy mud, 80 – Muddy sand, 100 – Sand, 110 – Gravelly mud, 115 – Gravelly sandy mud, 120 – Gravelly muddy sand, 130 – Gravelly sand, 150 – Muddy sandy gravel, 160 – Sandy gravel, 170 – Gravel. Mud, sandy mud, and muddy sand were reclassified as potential accumulation (1), the remainder as erosion/non-deposition (0) areas. The polygons were simplified with the Boundary Clean tool in ArcGIS. The potential accumulation areas were critically reviewed in the light of measured sedimentation rates and geological interpretations of sediment cores (de Haas et al., 1997 and references therein) and the dominant areas of erosion/net-deposition (1) and sediment accumulation (2) derived.

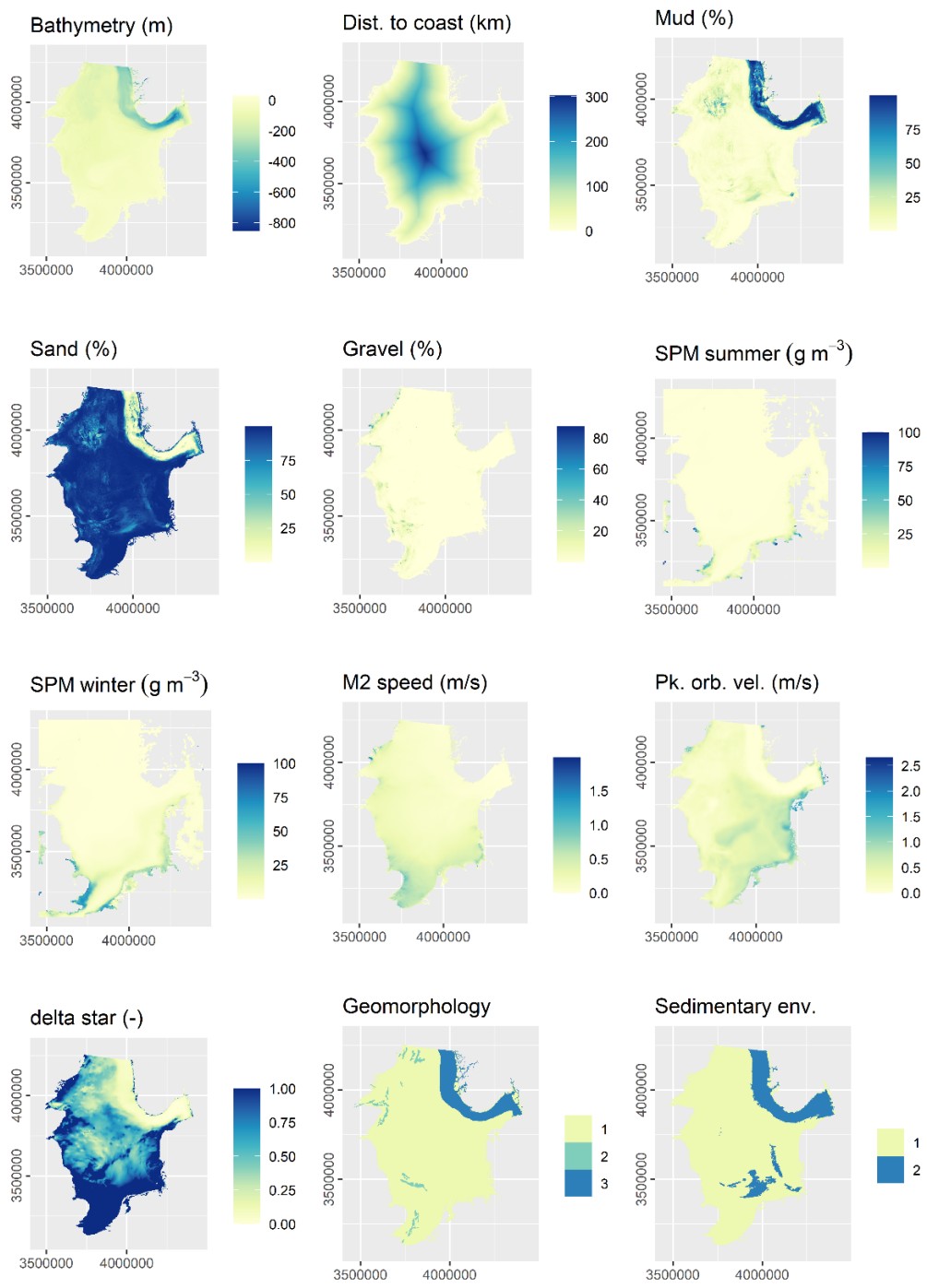

**Figure A2: Selected predictor variables of the sedimentation rate model. Geomorphology: 1 - Shelf, 2 – Shelf valley, 3 – Glacial trough. Sedimentary environment: 1 – Erosion/non-deposition, 2 - Accumulation.**

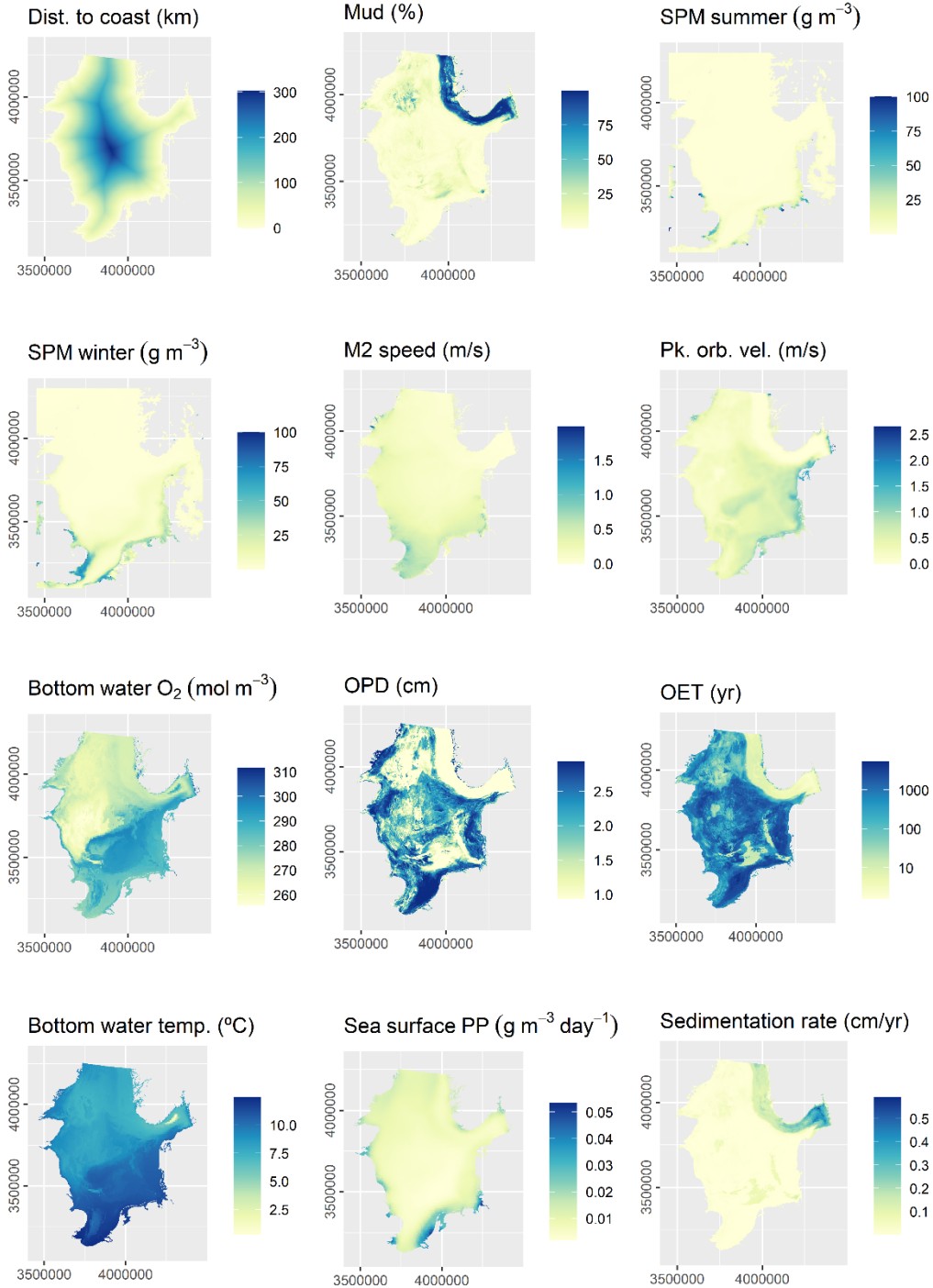


**Figure A3: Selected predictor variables of the OC density model. Bathymetry was also selected but is not shown here.**