# Peer review of "Organic carbon densities and accumulation rates in surface sediments of the North Sea and Skagerrak"

_Biogeosciences, 2020_

## Referee Comment (RC1) · Anonymous Referee #1 · 26 Oct 2020

General comments The study takes stock of organic carbon sequestered in surface sediments (0-10 cm) of the North Sea and Norwegian Trough/Skagerrak, and extrapolates sparse (and regionally extremely clustered) sedimentation rate records based on 210Pb to estimate annual organic carbon accumulation. Based on these existing data (analyses over the last decades), it applies sophisticated upscaling approaches and extensive error estimates, based on data from the EEZs of the UK and Norway. Using a suite of potential predictor variables (distance from shore, water depth, gridded sediment properties, numerical hydrodynamic model, etc), the main depocenter of the entire North Sea is (again) found to be the Skagerrak. Although this is hardly news (cited papers by Eisma, van Weering and de Haas and others), this is the first published, rigorous and statistically elaborate examination of C stocks and burial rates

in the entire shelf sea environment. What is new is the estimates of errors, and this is relevant as an incentive for further dedicated ground-truthing work. Although the manuscript offers essentially no surprises, because the resulting estimates of stocks and accumulation rates are close to previous estimates, the intrinsic motivation on the other hand is more interesting and potentially far-reaching. This is because the data are discussed in context of carbon accounting: A case is made for including substantial sedimentary carbon stocks in surface sediments, which greatly exceed other "blue carbon" stocks, into national carbon management accounts. This in turn raises some intriguing questions in a high-energy shelf sea environment such as the North Sea, where lateral transports of fine sediment fractions and associated organic carbon spiraling feeds accumulation remote from the production sites, and where erosion exhumes old carbon from a large, international catchment. How will this carbon sequestration be partitioned into national shares and accounted for? Will the credits go to the country that provides the storage space, or to the countries that provide nutrients for production of biological material? These accounting questions are clearly not at the focus here, but are highly relevant and offer potential for conflict. They should at least be pointed out. The manuscript is well structured, written and illustrated. It is very explicit on technical terms and concepts of data processing, which are probably beyond the expertise of most readers. I for one have to simply trust that the very elaborate approach is appropriate for the purpose, superior to more simple statistic procedures, and that results are correct. As stated above, the explicit elaboration on errors of the analysis is welcome, as are suggestions on how to reduce these errors.

Specific comments Some questions pertaining to the data base(s): Why have only data from the UK and Norway been analysed, although there must be significant stores of data from other countries? Have the authors checked the OSPAR data bases, or PANGEA for easily accessible data sets? To fill data gaps, the authors employed a technique of placing "pseudo observations" (Hengl et al., 2017 are cited as a reference). Did the authors just say: "We need a data point here – let's make it 0.1 wt.% organic carbon and 0 cm/yr sedimentation rate"? This needs to be explained in more

detail. Although not explicitly stated, the most important predictors emerging from the analysis are rather intuitive and straightforward (judging from Figure 6): bathymetry, oxygen penetration depth, and energy at the sea floor. Also, OC density and accumulation rates are included as predictors – but aren't these two the parameters to be estimated? The physical predictors are but a few of a long list of parameters that have gone into the analysis (in Table 1). Of these important (there is no measure of statistical impact given for any of them) predictors, bathymetry and energy at the seafloor are not independent (orbital velocities), and the way that oxygen penetration depths are calculated (as a function of mud content) makes one wonder, if they are not linked to bathymetry and energy at the seafloor as well. Instead, one would have expected that OC accumulation and OC standing stock are linked to grain size, for which spatially explicit data sets are available (e.g., Bockelmann et al., 2017) – was there no statistic relationship? If oxygen penetration depth is indeed a function of mud content, why is not mud content the logical predictor? What is the depth of oxygen penetration calculated by the empirical formula used here in the first place? This important piece of information is shown in Fig. 6, but eyeballing the box-and-whisker plot for the turnover zone in the figure and the profiles given in Lohse et al. (1996) suggest to me that the formula used here overestimates the oxygen penetration depth. This will not make much difference in the overall conclusion that not much carbon is stored anywhere except the Norwegian Trough and Skagerrak, but then oxygen penetration depth is probably not a good predictor of OCAR and OC density, because animals influence that depth. There are much more elaborate (model-based) and spatially contiguous estimates for oxygen penetration depths available at least in the southern North Sea (e.g., Luff and Moll, 2004; Pätsch et al., 2018, and probably more recent ones as well), and it might be advisable to use such data instead of the (apparently upublished) empirical relationships used that may be only regionally applicable. Finally, Zhang et al. (2019) highlight the important role of macrozoobenthos in recycling (and preserving) organic carbon at the sediment-water interface in much of the southern North Sea, an aspect not addressed in the present manuscript. An important issue of course is the origin

of OC, stated as an open question in the suggestions for future research. In general, the discussion on why some parts of the sea floor do and others do not accumulate organic carbon (section 6.2) implies that in-situ production and processing dictate OC density and OCAR. Resuspension and transport, and associated exposure to progressive degradation are not discussed. But why then do the hydrodynamic predictors (M2 current and orbital velocity) apparently have such a strong influence? Fine-grained sediment and associated OC buried in the Skagerrak and Norwegian Trough comprise input from rivers, atmospheric input, coastal and sea floor erosion, and primary production that feed into suspended and bed load transport of the North Sea. There are data sets on C/N ratios and even delta13C (see for example the thesis by de Haas, Geologica Ultraiectina 155, 1997), which indicate that about 20% is of terrigenous origin, and an unspecified source is erosion of older strata. Mineralisation in these muds only affects that (relatively small) portion supplied by production in the surface layer (see papers on oxygen consumption measurements by Lohse, Helder, Rysgaard etc., ). Finally, the manuscripts advocates and discusses at some length whether marine protected areas should be established to prevent accumulated organic carbon from being resuspended and remineralised. The manuscript cites an astounding estimate of potential damages arising from mineralisation caused by demersal fishing for the UK as support. Assuming that physical disturbance by demersal fishing enhances sedimentary carbon recycling, the logical sites for such MPAs and fisheries exclusion zones will be the accumulation areas – Skagerrak and Norwegian Trough. It would be interesting to know what the swept area ratio in these deep-water environments actually is and whether there have been monitoring activities (underwater video) in the fisheries sector to establish the extent of sediment reworking there. From the data on composition of OC in the depocenters, de Haas (1997; thesis) concludes that this OC is very recalcitrant – will it be further mineralised at all? Bakker and Helder (1993) showed that oxygen fluxes in the Skagerrak were not related to the total organic carbon (TOC) content of the sediments, and that apparently only fresh organic carbon from sinking primary production was mineralised. This suggests that the old carbon derived from

lateral transport across the large submarine catchment is unreactive. An interesting ancillary question is, whether (and if so, how much) a decline in (postulated) remineralision from artificial disturbance will negatively feed back on primary production and thus will reduce the amount of carbon supplied for burial.

Some additional remarks and comments keyed to line numbers: #75: this is only a temporary uptake # 106: erosion is an unquantified term and there is reason to believe that a substantial fraction of exhumed OC accumulates in recent depocenters

**122: which grain density was assumed? #136: water depth = distance to shore? Sedimentation rate according to Müller and Suess is relevant only for deep-sea sediments and vertical transport**

**140 ff: What use are geomorphology features beyond water depth and orbital velocities? Why not use numerical values for grain sizes (Bockelmann et al., 2017)?**

**145: There are no dated cores in several of these areas (judging from a comparison of Figures 1 and 2)**

**148: LSR is 0 in most of the area, and besides the oxygen exposure time appears to be not relevant.**

**171: Which are these? Include information of statistical weights/relevance as predictor in Table 1**

**216 ff: Are these the environmental variables explaining most of the variance? O2 penetration depth was estimated by mud content - is that independent of OC density? Table 1 lists many predictor variables, most of which apparently are not crucial. On the other hand, sedimentation rate is a crucial parameters, but how are grain size/Folk parameters etc. linked to sedimentation rate?**

**221: Show loadings in a figure or table**

**243: The mode of transport by OC spiraling is important here - after what time does**

the material from production sites arrive at the depocenter, and how is the material reworked on the way?

**247: Are these parameters the dominant discriminators?**

**291 ff: This raises the interesting question on who reports -producer or storage provider?**

**297: The difference is that OC in fjord sediments probably is autochthonous (or from land), whereas Norwegian Trough collects OC from a large submarine area**

**312: I assume that these characteristic parameters are the ones shown in Figure 6 (bathymetry, oxygen penetration depth derived from mud content, energy at the sea floor from M2 and orbital velocity)? See comments above.**

**393: There is always room for improving the data base and suggestions are certainly valid. But when most samples are sands with <0.2 wt.% OC, a standardisation of methods (which effectively has been done in the past on various occasions for the analytical steps) will not change the general conclusions. In the de Haas thesis available on the internet (Mededelingen van de Faculteit Aardwetenschappen Universiteit Utrecht No. 155: Transport, preservation and accumulation of organic carbon in the North Sea), there is information on the delta 13C of OC and C/N ratios. #395: Again - who is credited for sequestration of allochthonous carbon in the Norwegian trench? Those who produce the carbon (in their EEZ′s), or those who store it in their EEZ′s? Productivity is highest in the southern, non-depositional sectors of the North Sea. The best way to enhance sequestration potential is to increase productivity – probably by enhancing eutrophication.**

Table 1: Add a measure of statistical weight Table 2: Reference needed in the caption Fig. 1: The upper bound of the color scale (51 m) appears to wrong. How were depositional areas selected? What is the diagonal white line from Denmark to the southern tip of Norway? Fig. 6: Are the parameters presented here as box-and-whisker

plots the ones that are dominant, or why were they chosen (see comment above)?
Fig.

---

## Referee Comment (RC2) · Anonymous Referee #2 · 29 Oct 2020

1. General Comments The authors of the manuscript "Organic carbon in surface sediments of the North Sea and Skagerrak" have presented a spatial modelling framework to predict and map the spatial distribution of surficial organic carbon (OC) densities and organic carbon accumulation rates (OCAR) in two regional seas in the North East Atlantic. The results are aligned with previous studies looking at (spatial distributions of) OC in the North Sea (de Haas et al., 1997 & 2002; Legge et al., 2020, Smeaton et al., 2020). However, this study provides novelty with its spatial approach to mapping OCAR (with associated uncertainty) and is a welcome contribution to net sediment accumulation and depocentre research. On this point the title of the manuscript could be updated to include OCAR for clarity. An interesting element to this study is the identification of the Norwegian Trough as being a highly effective accumulator and store for OC (although the authors themselves note on line 94 that this has been previously observed in a study over 20 years ago), and the authors suggest, as effective as the nearby fjordic environments, which is an unusual finding for continental shelf sediments. The break-down of regions into different 'carbon-processing zones' is a novel concept in mapping, however, more thought is required in the discussion as to the physical and biogeochemical processes that transport and cycle carbon on the shelf – and acknowledging the differences between OC accumulation and OC burial (e.g. Zonneveld et al., 2009). The provision of the R Markdown files is welcome and useful for traceability of results as is the provision of the raw data used in the models. An additional table of the outputs used to make final OC stock estimates would be useful (for those who can't/don't use R). The paper is well-written and concise but some more references to the supplementary data and figures would be useful as well as to support some statements (Detail provided below). There are some elements of this manuscript that could be improved, and I have provided detailed comments and suggestions in 'Specific Comments' and 'Technical Corrections' below. They relate primarily to: providing more detail about the data and how they have been selected/processed; how the final results have been calculated (a table would be useful to break down the component outputs from the model); further development of some of the processes mentioned (C-cycling, transport, C-accounting) and core arguments and in the discussion to realise the impact of these findings and; re-wording some paragraphs to help with comprehension and clarity. The aims of the paper are important and relevant, although I question the usefulness of comparing these OC stocks to other stocks; for instance, coastal ecosystems have different mechanisms for sequestering carbon and are spatially limited. Soils are more comparable by area, however presumably there are much more data available due to ease of sampling and therefore lower uncertainties? Uncertainty estimates in this paper for the sedimentation and OCAR are quite high (same order of magnitude) generally, and I wondered why they were highest in areas with a higher density of data points? (This could be a misunderstanding on my part of the model, but detail would promote clarity!) I think this paper needs to acknowledge the differences

between accumulation rates and burial rates – for instance in section 6.3, it is stated that "zones of OC burial" have been identified, however there was no investigation into how the OC density varied with depth to comment on how effective this site is for burial, and this is an important distinction to make. Data for the model are limited, with few to no datapoints over large areas of the North Sea and large assumptions are made. Further details would be welcomed relating to data selection, model outputs, interpretation of RSME and variance in the results, and some assumptions could be strengthen by links to the literature (e.g. OC change with depth; oxygen penetration as a function of mud). There is noticeably little discussion of the effect of sediment type on OC which has been shown to be a significant predictor of OC.

2. Specific Comments I have provided comments that address individual scientific question or issues below. The comments are broken down by section, and line numbers are given to address specific issues. #Introduction It sets the scene well but more clarity needs to be give as to how this study is novel compared to other predictive spatial models for OC stocks (e.g. is it due to a new framework, a different location being studies, or is it about calculating accumulation rates?). The link between OC and sediment type isn't clear, although a focus is made of cohesive sediments. Can the authors expand on what these are and why are the more relevant to OC? Some more detail could be included about the benefits of random forest modelling as a rationale for why this method 'appears' to have been chosen in recent modelling studies. The text from lines 48 – 57 could be strengthened. Why should marine carbon stocks be accounted for and what kinds of damage are possible as a result of disturbance. #Line 27 – Can the authors suggest what other differences might account for large differences in global stock estimates? #Line 39 – Misleading - suggests the authors will look at burial rates as well as accumulation. #Line 44 – Suggest the authors make reference to these fjord studies coming from the UK (other fjordic studies are available if the authors wanted a more global perspective in this argument). Is this study trying to improve the North Sea estimates specifically or estimates generally? (Line 41) #Line 47 – The inclusion of Namibia is unexpected in this comparison. What is the

relevance? #Line 53 – Suggest removing 'projects' (replace with strategy?) – I don't think stocks themselves can be used to mitigate against GG emissions. #Line 55 – Can the authors provide some detail about the kinds of damage attributed to carbon release? #Line 59 – To strengthen the idea of using MPAs, can the authors provide some detail as to how MPAs have been used to protect BC carbon storage? #Line 74 – A note to reflect on the likelihood of MPAs (especially on this scale) being developed to protect the seabed against demersal fishing – this isn't a straight-forward decision. #Line 79 – It is not clear how linking to an area most heavily impacted by human activities is ideal for understanding accumulation rates – the study isn't necessarily looking at the effects of human activities on accumulation rates. #Data Generally, some more detail is requested for the final datasets used (there are large areas of the North sea with no data – do they not exist?), some of the assumptions made on sedimentation rates and the criteria used to assign accumulative areas. Are figures or supplementary datasets available for the oxygen penetration depth and oxygen exposure time? I'm not clear from the text what form these data take – continuous raster layers? Oxygen exposure time is calculated using the sedimentation rate which is modelled within this study – so the uncertainties will be carried across presumably. Are the Haas data reliable? Some more detail on why certain values were changed and the criteria used to make these decisions would be useful. #Line 108 – Can the authors elaborate on what pseudo-observations are and if they are comparable? #Line 111 – What was it about the 210-Pb profiles that made the authors reject some data? #Line 116 – Suggest making a reference to Supp Data Table and provide some more detail in the text for these data. Where have the OC measurements come from? How many etc. #Line 124 – Refer to Supp Data Table for reference. #Line 143 – Suggest including a figure to show the Folk classes of the area and the 'cleaned' boundaries. #Line 145 – What criteria were used to decide whether an area was potentially accumulative or not? #Line 147 – Can the authors describe generally what the relationship between measured oxygen depth and mud content is expected to be? Does oxygen penetrate more or less in mud? What is the relationship to cohesive sediments? #Method The

use of the QRF Random Forest model is well justified, and the methods are clear / concise. Some detail on what the different types of error / variance generated mean would be useful and how this differs from the coefficient of determination. #Line 198 - Would be useful to provide a conversion factor to OC stocks from other studies referenced in this study e.g. Tg - Mt / Tmol and that use different units. This would make inter-study comparisons easier / more transparent. #Line 204 - Somewhere it should be noted that there is a difference between carbon accumulation rates and burial rates (i.e. just because carbon is accumulating, doesn't automatically mean it is being buried in the same amounts) #Line 221 - Specifically what were these variables that accounted for 95.5% of the variance? #Results Concise reporting – although it is not entirely clear how to interpret / use the RMSE and Explained Variance values. A table showing how the final results have been derived would be useful – can the model output results at specific stages? A breakdown of the average sedimentation / OC density and OCAR results by the three regions would be useful. It is not clear to me why there is higher uncertainty in higher sedimentation rates which is also where there is a higher density of data points. The results section might not be the correct section to answer this but do the authors have any insights into why there is a much higher proportion of OC accumulating (87%) in the Norwegian Trough than the proportion stored here (25.9%) – Is there high turnover here? The discussion mentions several characteristics of this area which enhance preservation of OC. #Discussion #Relevance – This section can be strengthened. Perhaps the section needs to be re-titled to "Context". There are many assumptions made (for instance how OC changes with depth), which increase the uncertainty in the scaled-up estimates (making it less useful for improved carbon stock accounts). The discussion on reporting uncertainties could reflect on how to improve uncertainties. The authors argue that their uncertainty estimates are robust because they are based on soil OC mapping studies which, will be different to the marine realm because sampling is easier and there are different predictor variables influencing OC distributions presumably. The comparison of shelf sediment stocks to coastal "blue carbon" doesn't acknowledge the differences between the ecosystems

e.g. that coastal habitats are spatially limited to the intertidal zone, have a much smaller areal coverage and has a different mechanism in terms of carbon sequestration. The argument for the Norwegian Trough as a unique and highly effective zone of carbon accumulation (if this is what the authors are trying to argue) needs to be re-worked for emphasis – it gets lost by the introduction of Scottish and Irish fjords. #Zones of OC processing at the seafloor – The first paragraph is too reflective and needs a few more references for statements. It isn't clear how initial studies of OC cycling on the shelf led to the notion of rapidly accumulating coastal sediments? The authors provide a useful summary of environmental seafloor processes to explain oxygen dynamics. Lines 339-342 are unclear that the characteristics listed are for sediment properties that influence OC cycling. Needs a little re-working. #Implications for management – This section is currently too vague. Although the implications of refining zones due to OC processing is an interesting concept and potentially a useful way of simplifying areas for management, the scales discussed for MPAs are probably too large to be effective or manageable. Natural disturbance hasn't been acknowledged.. #Suggestions for Future Research – More detail is required around further data collection – the goal of data collection needs to be elaborated and more thought into specifically what data would be useful / beneficial to collect to answer questions relating to carbon stocks. The sampling design examples are very technical - who might undertake this enormous task? Some detail about existing data stores would be useful for reader understanding that national and pan-European datacentres do exist. I think some further discussion on the ideas behind 'source of OC' – and why this might be relevant to further study in terms of thinking about climate mitigation – is needed. The authors presume this is common knowledge. #Line 260 - A figure would be useful to put the 'global continental shelf' in the context of the global seafloor (and then the two regional seas into context as well). #Line 264 – The assumption is very vague - are there any studies that provide an estimate of how OC stock changes with depth to get a narrower estimate? #Line 270 - Where does 58% OC stock uncertainty come from? Line 233? (Explained variance?) #Line 272 – The comparison to lower uncertainty values from

local studies could be further developed. #Line 274 - Is this a good comparison? Does soil OM have similar predictor variables (e.g. current speed?) Soils are presumably easier to sample as well and therefore have a better spatial range of samples. Some further development of the argument would be helpful. #Line 282 - Coastal habitats are limited spatially by depth and limited to coastlines - generally intertidal zone which is not considered the continental shelf. Can the authors provide an area estimate for these coastal habitats to provide context for the OC-stock values reported? How do the OC densities compare when normalised to area? #Line 293 - The word project is ambiguous and implies that sediments can be managed to increase sequestration of $CO2$. The link between greenhouse gases and OC found in sediments is not made. What are the implications for accounting for these stocks? National inventory numbers would increase - but how can this be useful of greenhouse gas reporting? #Lines 300 – 304 - Suggest rearranging - I think the authors are trying to say that the Norwegian trough could be an OC accumulation zone unique even to fjord environments because it is apparently not heterogeneous? #Line 303 – Reference / figure to back-up that the Norwegian trough has homogenous sediment? #Line 351 – "Potential zones of OC burial" - there was no investigation into how the OC density varied with depth to comment on how effective this site is for burial – this should be removed. #Line 375 – Further discussion about what new samples are being recommended for collection. To collect what specifically? grain size? Carbon measurements? to what depth? In-situ oxygen / current data / sedimentations rates?? What are the questions / gaps to inform what data are required? #Line 378 – Could you elaborate on how economic benefits can be achieved? #Line 380 – What type of baseline dataset? #Line 388 - Agreed - however there are national sampling programmes that have standardised protocols – do these need to be advertised' to the research community / or informed by the research community? #Line 391 – Such facilities do exist – e.g. ICES. #Line 396 - How likely is there to be terrigenous OM in shelf sediments? Any studies that have looked at this?

3. Technical Corrections Comments are provided with specific line references for consideration: #Line 7 – Suggest re-wording; Sediments don't protect the seabed from disturbance, Sediments can store carbon, provided left undisturbed (from anthropogenic activity). #Line 10 – Inclusion of 'us' between 'allows to'. #Line 16 – Suggest updating 'on par' with 'comparable'. #Line 30 – Suggest replacing 'were' with 'have been'. #Line 31 - Suggest replacing 'did not include' with 'have not included'. #Line 33 – Reference for importance of continental margins in OC cycling – and important in what way? #Line 37 – Use of the word 'appear' without suggesting why this might be. What are the advantages of machine learning over geostatistical approach? #Line 41 – Suggest replacing 'point of view' with 'perspective'. #Line 42 – Reference for the inclusion of 'potentially macroalgae' in the BC definition. #Line 63 – Add 'The'. #Line 64 – Suggestion for consideration. Is 'fertilization' the right term? to fertilise means to stimulate productivity - this would reduce OC presumably. Is enrichment a better term? #Line 66 – Suggest including 'sediment' between 'deeper layers'. #Line 69 – Change 'expectable' to 'expected'. #Line 78 – suggest replacing 'it is one of the regional seas' with 'they are the' for comprehension. Regional setting – Figure 1 - Request to add the labels for the two regional seas on figure 1 location map. #Line 99 – Suggest re-wording 'generally deepening from south to north'. Specific depths? #Lines 117, 118, 120 – Use of the word concentration is incorrect. Update to content (mass per unit mass) – See Flemming & Delafontaine, 2000. #Line 133 – suggest adding 'relevance to OC'. #Line 139 – Suggest addition of appropriate reference to reinforce-up this statement. #Line 155 – Suggest replacing 'target' with 'response' to keep the terms consistent. #Line 158 – Inclusion of the word 'us'. #Line 183 - Would be useful to include a sentence describing what the RMSE explains (and the difference between this and the MSE in the context of the model performance) #Line 273 - Suggest replacing 'how' with 'in which'. #Line 283 - 284 – Sentence doesn't make sense compare to preceding sentence. #Line 285 – Does 'collectively' mean 'global'? #Line 315 – References to initial process studies? #Line 316 – How did one lead to the other? #Line 339 – 341 – Sentence isn't well constructed or complete. #Line 342 – Suggest inclusion of 'a' burial zone. #Line 350 – Suggest replacing 'on par' with 'comparable'. #Line 351 – Suggest replacing 'act

differently' with 'have different roles'? ('act' suggests it is a conscious action - not a by-product of location and physical environment #Line 354 – I don't understand the point about 'total annual rate in the North Sea'. #Line 370 – Suggest adding 'However' at start of sentence. #Line 382 - Relative importance on what? I assume OC but this isn't explicit. #Line 405 – Suggest replacing 'on par' with comparable.

———————————————

---

## Referee Comment (RC3) · Anonymous Referee #3 · 11 Nov 2020

General comments: The authors successfully present a combined modelling framework of sedimentation rates and Organic Carbon (OC) densities, used to determine the spatial variability of OC in the North Sea and Skagerrak regions. The methods are outlined clearly and the results are presented in a way that is easy to understand. Although the uncertainties are unfortunately quite high, the work presented in this manuscript represents a valuable contribution to the field that should be published. Specific comments: Lines 108-109: Give a short definition of "pseudo-observations" in the context of this work. Lines 111-112: Define how many are meant by "Some of the sedimentation rate values...", does this refer to the four values that are amended later in the same sentence or are these four a subset of the "some"? If it's a subset, the selection process should be explained. Line 131 (Figure 2): There seem to be

no OC measurements in the Elbe Paleo valley region (Region 2), if this is the case it should be explicitly mentioned. Line 145: "critically reviewed and removed if they were not deemed accumulative" an explanation on the selection/removal criteria should be added here. Lines 263-264: "It is therefore safe to assume that the sediment slice between 5 and 10 cm will contain between 0 % and 100 % of the OC stock of the upper 5 cm." It is generally safe to assume that anything contains between 0% and 100% of anything, so this sentence is either unnecessary, or should be reworded in a way that makes more sense. Lines 339-341: "Lack of advective oxidation [. . .] and relatively high sedimentation rates." The wording of this sentence is unclear and should be revised. Figures: Very clear and easy to understand, good work.

―――――――――――――――――――――

---

## Short Comment (SC1) · 18 Nov 2020

Line 87: In the Regional setting or Data sections it would be worthwhile stating why the boundaries of the study site were selected. I presume this was due to the overlapping extent of predictor variables listed in Table 1. However, it is unusual that the focus does not cover the complete extent of any of the countries EEZ presented in this study (Table 2), because had it done so this would improve the impact of the current piece of work. As the discussion encourages further research of this kind (Section 6.4 and the publishing of R scripts), it would be worth clarifying whether similar predictor data are available, or whether these too would need to be generated first. If on the other hand it was due to available sample data or how far the authors felt they could extrapolate the models then this would also be of interest to future scientists doing similar work.

[Figure]

Line 190: There appears to be an error in the calculation of VE and rˆ2. While rˆ2 is also termed the coefficient of determination, it is my understanding that the VE and rˆ2 are the same metric. Therefore, I was surprised to see such different results reported in Line 226 and 233. Looking at the R Markdown code to understand how these two values have been calculated I see that calculation of VE contains the test predictions within the denominator in:

validation[i, 3] <- 1-(mse(df$test.SedRate, df$test.pred)/var(df$test.SedRate, df$test.pred))

As VE is calculated as the unexplained variation over the total variation its not clear to me why the denominator in your calculation has the test set predictions. Suggest checking your formulas to ensure the values presented are correct. Its also not clear to me whether both metrics are required or tell a story that is not captured by rˆ2. So you may wish to present rˆ2 only.

Line 253: Starting the discussion by referring to the R Markdown code and seemed a little out of place. As the results of this study are a valuable contribution to the field of Blue Carbon which is rapidly gaining interest to develop policies in various European governments, this focus on encouraging use of the scripts may be less interesting to the reader. Authors may consider moving this to section '6.4 Suggestions for future research' and instead focussing on the main findings of the study.

Line 259: Similar to above comment. Section '6.1 Relevance' starts with a recap of other research and not the findings of this current study. Authors should consider whether to lead with what this study has shown and then put that into context of other work to show the relevance.

Line 260: Does Harris et al 2014 need to be referenced here? Suggest deleting. Also, Lee et al. (2019) present maps of uncertainty for their estimates of OC. Relative to the assumptions presented in line 263-265 (total stocks vary between 12.1-24.2 Pg C), should the Lee et al. uncertainty be accounted for in this estimation, or are they at a

much smaller magnitude? As the uncertainty map in Lee et al does seem to show that uncertainty is also concentrated around the continental shelf.

Line 264: 'between 0% and 100%'. I am struggling to follow what is being said in this sentence. Are you simply stating that the OC in 5-10cm does not exceed that in 0-5cm? As that was already stated in the previous two sentences. Unless i am missing some subtle difference.

Line 284: Is this sentences stating that the shelf sediments of the European Continental Shelf are an order of magnitude greater than coastal habitats, based solely on the calculations for the North Sea/Skagerrak? Or that is the reference to 'smaller area' comparing the area covered by the North Sea/Skagerrak relative to the area covered by coastal habitats? I assume the first as no area figures have been presented for comparison of the latter. Consider rephrasing for clarity.

---

## Author Comment (AC1) · 9 Dec 2020

Reply to RC1

Reviewer 1 attests that this study "is the first published, rigorous and statistically elaborate examination of C stocks and burial rates in the entire shelf sea environment. What is new is the estimates of errors, and this is relevant as an incentive for further dedicated ground-truthing work."

General comments:

Comment: How will this carbon sequestration be partitioned into national shares and accounted for? Will the credits go to the country that provides the storage space, or to the countries that provide nutrients for production of biological material? These accounting questions are clearly not at the focus here, but are highly relevant and offer potential for conflict. They should at least be pointed out.

Reply: As stated by the reviewer, the above questions are not the focus of our study. However, we briefly mention potential implications in the introduction now.

Action: The introduction was updated including a reference to a recent paper on that topic:

[…] Consequently, the question has been raised whether those stocks should be considered as part of national carbon accounting and potential greenhouse gas mitigation projects and subject to management against human-induced disturbance (Avelar et al., 2017). The socio-economic importance of marine carbon storage has recently been assessed in a scenario analysis of increased human and climate pressures over a 25-year period. It was estimated that damage costs of up to $12.5 billion from carbon release linked to disturbance of coastal and shelf sea sediment carbon stores could arise in the United Kingdom (Luisetti et al., 2019). *However, the transboundary nature of carbon flows in the marine environment poses significant challenges for carbon accounting and requires new guidance and governance frameworks to manage these stocks (Luisetti et al., 2020).*

Specific comments:

Comment: Why have only data from the UK and Norway been analysed, although there must be significant stores of data from other countries? Have the authors checked the OSPAR data bases, or PANGEA for easily accessible data sets?

Reply: The data on sedimentation rates were mainly sourced from the EMODnet-Geology data portal. We are confident that this is the most comprehensive collection of measured sedimentation rates for European seas. Any geographic bias is due to the locations of sediment accumulation basins, such as the Norwegian Trough. To address this bias, data from de Haas et al. (1997) were also included to cover areas of low or no net sedimentation. These were collected on a widely spaced grid across the entire North Sea. Regarding organic carbon (OC), we decided to predict OC densities (kg m$^{-3}$) instead of OC contents (%). As pointed out in section 3.1.2, this has two advantages from a methodological point of view: First, there is no need to transform the response variable as would be necessary in the case of OC contents reported as weight-% or fractions. Second, only one model instead of two needs to be fitted. Especially the second point is relevant here, as fitting two models (one for OC and one for dry bulk density or porosity) would likely increase the uncertainty of the predictions. However, this means that only data sets that report OC contents together with dry bulk density/porosity could be utilised. The final data were carefully selected after screening various data bases and other sources including PANGAEA and ICES.

Action: Section 3.1.2 was rewritten:

Previous studies have predicted OC concentrations and sediment porosity separately to calculate OC stocks (Diesing et al., 2017; Lee et al., 2019; Wilson et al., 2018). Here, we first calculate OC density from concurrent measurements of OC concentrations and sediment dry bulk densities or porosities. This has two advantages: First, there is no need to transform the response variable as would be necessary in the case of OC concentrations reported as weight-% or fractions. Second, only one model instead of two needs to be fitted. *This is advantageous as fitting two models would likely increase the uncertainty of the predictions. Initially, a wide range of data sources were accessed. Ultimately, 373 samples fulfilled the criterion of providing OC content and dry bulk density/porosity measured on the same sample. These samples were collected and measured by the Geological Survey of Norway, the Centre for Environment, Fisheries and Aquaculture Science, Bakker and Helder (1993) and de Haas et al. (1997). The full dataset used for subsequent modelling is shown in Figure 2 and provided as Supplementary Data Table 2.*

Comment: To fill data gaps, the authors employed a technique of placing "pseudo observations" (Hengl et al., 2017 are cited as a reference). Did the authors just say: "We need a data point here – let0s make it 0.1 wt.% organic carbon and 0 cm/yr sedimentation rate"? This needs to be explained in more detail.

Reply: We believe that it is clear from the manuscript that pseudo-observations were only used in the case of sedimentation rates. No pseudo-observations were employed to model OC density. As explained in section 3.1.1, it was necessary to place pseudo-observations when modelling sedimentation rates due to a strong bias of samples towards accumulation areas. However, large areas of the North Sea are non-depositional or erosional in nature and only those samples from de Haas et al. (1997), less than 20 in total, were taken in these areas. We therefore resorted to pseudo-observations, a practice that has been applied in the past (Hengl et al., 2017). Pseudo-observations were only placed in areas where we were confident that sedimentation rates were 0 cm/yr. The placement was conducted in a random way to avoid human bias. We understand that this method might sound arbitrary at first and that many marine scientists might not be familiar with the concept. As suggested by the reviewer, we are now giving more details in section 3.1.1.

Action: Section 3.1.1 was amended to give a more detailed explanation on pseudo-observations:

The reported sedimentation rate data focussed on accumulation areas like the Norwegian Trough (Figure 2). However, to be able to spatially predict sedimentation rates across the study site it is necessary to include data from areas of erosion and non-deposition, which predominate in the North Sea. Therefore, the data of de Haas et al. (1997) were also included. This provided less than 20 data points of zero net-sedimentation, which was still deemed insufficient. Additionally, pseudo-observations (Hengl et al., 2017) were also included. *Pseudo-observations are 'virtual' samples that are placed in undersampled areas and for which the value of the response variable can be assumed with high certainty. Hengl et al. (2017) cite 0 % soil OC in the top 2 m of active sand dunes as an example. Mitchell et al. (in review) placed pseudo-samples in areas of bedrock outcropping at the seabed when predicting sedimentation rates in the Baltic Sea. The placement of pseudo-observations was restricted to* areas of erosion and non-deposition (based on the sedimentary environment layer, as described in chapter 3.2)*, for which*  a sedimentation rate of 0 cm yr$^{-1}$ could be assumed. *The pseudo-observations were placed randomly to avoid human bias.* [...]

Comment: Although not explicitly stated, the most important predictors emerging from the analysis are rather intuitive and straightforward (judging from Figure 6): bathymetry, oxygen penetration depth, and energy at the sea floor. Also, OC density and accumulation rates are included as predictors – but aren't these two the parameters to be estimated?

Reply: We are sorry for the confusion, but the parameters in Figure 6 are not the predictors. Instead, Figure 6 shows the variables that were employed in the unsupervised classification (section 4.4). We did not state explicitly in the text, which predictor variables were chosen for modelling. Instead this information can only be found in the Supplements S1 and S2.

Action: To avoid confusion, we now provide information on the selected predictors in the results section.

Comment: The physical predictors are but a few of a long list of parameters that have gone into the analysis (in Table 1). Of these important (there is no measure of statistical impact given for any of them) predictors, bathymetry and energy at the seafloor are not independent (orbital velocities), and the way that oxygen penetration depths are calculated (as a function of mud content) makes one wonder, if they are not linked to bathymetry and energy at the seafloor as well. Instead, one would have expected that OC accumulation and OC standing stock are linked to grain size, for which spatially explicit data sets are available (e.g., Bockelmann et al., 2017) – was there no statistic relationship?

Reply: Again, the information on predictor variable importance is "hidden" in the supplements. We did not intend to discuss the importance of predictors in this manuscript, as a better understanding of the links between predictor and response variables was not the focus of the work. However, it seems that it might be prudent to include such information in the main text. Also, we did not model OC accumulation and OC standing stock; both were calculated from OC densities (see sections 4.2 and 4.3). The OC density model did include mud content as an important predictor, although mud was maybe less important than could have been expected from previous work (e.g. Diesing et al., 2017). This might be attributable to the fact that we predicted OC densities rather than OC content. Muds typically have high OC content but also low dry bulk densities. Consequently, it can be expected that the relationship between OC density and mud content is weaker than between OC content and mud content.

Action: We now include a figure showing the selected predictors and their importance scores.

Comment: If oxygen penetration depth is indeed a function of mud content, why is not mud content the logical predictor? What is the depth of oxygen penetration calculated by the empirical formula used here in the first place? This important piece of information is shown in Fig. 6, but eyeballing the box-and-whisker plot for the turnover zone in the figure and the profiles given in Lohse et al. (1996) suggest to me that the formula used here overestimates the oxygen penetration depth. This will not make much difference in the overall conclusion that not much carbon is stored anywhere except the Norwegian Trough and Skagerrak, but then oxygen penetration depth is probably not a good predictor of OCAR and OC density, because animals influence that depth. There are much more elaborate (model-based) and spatially contiguous estimates for oxygen penetration depths available at least in the southern North Sea (e.g., Luff and Moll, 2004; Pätsch et al., 2018, and probably more

recent ones as well), and it might be advisable to use such data instead of the (apparently upublished) empirical relationships used that may be only regionally applicable.

Reply: Both mud content and oxygen penetration depth were selected as predictors in the OC density model. All predictors, including oxygen penetration depth, are now included in additional figures. This will give a better overview of the spatial variation of predictors, including oxygen penetration depth. We assume the reviewer is referring to Lohse et al. (1996). In their Fig. 4, oxygen penetration depths > 4 cm were measured on Dogger Bank and of ≈ 2.7 cm off Esbjerg. Painting et al. (2013) report measured mean oxygen penetration depths of 4.8 ± 2.4 cm in the Southern Bight. Likewise, Hicks et al. (2017) report an oxygen penetration depth of 4.6 cm at a permeable sediment site in the Celtic Sea. Taken together, this leads us to believe that our values of up to 2.9 cm in the turnover zone are reasonable and no overestimates. Using the suggested study results (Luff and Moll, 2004; Pätsch et al., 2018) as input data is problematic because they might not cover the whole extent of the study area (as mentioned by the reviewer) and outputs might not be freely available (based on experience), but this is difficult to ascertain without complete references. We therefore believe that using oxygen penetration depth estimated from mud content is the preferred option under the given circumstances.

Action: We now include additional figures showing the selected predictor variables.

Comment: Finally, Zhang et al. (2019) highlight the important role of macrozoobenthos in recycling (and preserving) organic carbon at the sediment-water interface in much of the southern North Sea, an aspect not addressed in the present manuscript.

Reply: We assume the reviewer is referring to Zhang et al. (2019). This is certainly an interesting paper dealing with related aspects of OC modulation by macrobenthos.

Action: We have updated the text in section 6.2 to briefly mention the importance of bioturbation for OC preservation.

Conversely, the seabed of the Norwegian Trough is characterised by water depths in excess of 200 m and experiences very subdued wave and current agitation. Fluid transport in the sediment is therefore driven by molecular diffusion, mediated by bioturbation. *Bioturbation contributes to a balance in the sedimentary OC budget by transporting labile OC to deeper horizons where degradation efficiency is lower (Zhang et al., 2019).* Lack of advective oxidation (Huettel et al., 2014; Huettel and Rusch, 2000) combines with mineral protection (Hedges and Keil, 1995; Hemingway et al., 2019; Keil and Hedges, 1993; Mayer, 1994) and short oxygen exposure times (Hartnett et al., 1998) due to shallow oxygen penetration depths and relatively high sedimentation rates. Collectively, this leads to high OC densities and accumulation rates. This zone might be termed burial zone according to Huettel and Rusch (2000).

Comment: An important issue of course is the origin of OC, stated as an open question in the suggestions for future research. In general, the discussion on why some parts of the sea floor do and others do not accumulate organic carbon (section 6.2) implies that in-situ production and processing dictate OC density and OCAR. Resuspension and transport, and associated exposure to progressive degradation are not discussed. But why then do the hydrodynamic predictors (M2 current and orbital velocity) apparently have such a strong influence? Fine-grained sediment and associated OC buried in the Skagerrak and Norwegian Trough comprise input from rivers, atmospheric input, coastal and sea

floor erosion, and primary production that feed into suspended and bed load transport of the North Sea. There are data sets on C/N ratios and even delta13C (see for example the thesis by de Haas, Geologica Ultraiectina 155, 1997), which indicate that about 20% is of terrigenous origin, and an unspecified source is erosion of older strata. Mineralisation in these muds only affects that (relatively small) portion supplied by production in the surface layer (see papers on oxygen consumption measurements by Lohse, Helder, Rysgaard etc., ).

Reply: Hydrodynamics were identified as relevant predictors; however, other predictors were more important. These were, in decreasing order of importance, bathymetry, sedimentation rate, mean bottom water temperature, oxygen exposure time and mud content. These are in good agreement with the expectations, as stated in section 3.2. It should be noted that Figure 6 was derived by an unsupervised classification approach (section 4.4). To carry out this regionalisation of the North Sea regarding processing of OC at the seafloor we chose the six variables displayed in Figure 6. Hydrodynamics (current speed and wave orbital velocity) were included as they play a crucial role in the rapid degradation of OC in permeable sediments (Huettel et al., 2014). Accumulative areas, on the other hand, are characterised by weak hydrodynamics, which favours the deposition of fine-grained sediments. Porewater transport is diffusive and hence much slower than in permeable sediments with advective transport. Together with mineral protection and short oxygen exposure times after sedimentation, this leads to increased OC densities and accumulation rates. We concede that the aspect of resuspension and transport of OC prior to deposition in the main depocentre, the Norwegian Trough, has been left out in our discussion of zones of OC processing (section 6.2.). This has now been rectified. We agree that it would be desirable to address the question of the origin and sources of OC deposited in the North Sea and Skagerrak. We think, however, that a detailed re-analysis of C/N ratios and $\delta^{13}$C values is beyond the scope of this study. Also, de Haas (1997) states in his thesis: "Variations in $\delta^{13}C_{org}$ in Norwegian Channel sediments cannot be used to explain variations in $C_{org}$ contents as a result of differences in source of the organic matter." Unfortunately, we were unable to ascertain which publications from "Lohse, Helder, Rysgaard etc." the reviewer was referring to.

Action: The discussion of the "burial zone" in section 6.2 was updated to reflect the origin of the OC deposited there:

*De Haas and van Weering* (1997) *estimated that only 10 % of the OC deposited in the Norwegian Trough is derived from local primary production and the remainder originates from other sources. A large part of this allochthonous OC is transported into the Norwegian Trough along the Dutch, German and Danish coasts by an anti-clockwise residual circulation* (de Haas et al., 2002). *This transport is thought to be intermittent, with the rate of transport dependent on the strength of wind-induced waves and currents (de Haas and van Weering, 1997). The OC being deposited in the Norwegian Trough is mostly refractory, as it has undergone several erosion-transport-deposition cycles prior to final deposition (de Haas et al., 2002).*

Comment: Finally, the manuscripts advocates and discusses at some length whether marine protected areas should be established to prevent accumulated organic carbon from being resuspended and remineralised. The manuscript cites an astounding estimate of potential damages arising from mineralisation caused by demersal fishing for the UK as support. Assuming that physical disturbance by demersal fishing enhances sedimentary carbon recycling, the logical sites for such MPAs and fisheries exclusion zones will be the accumulation areas – Skagerrak and Norwegian Trough. It would be interesting to know what the swept area ratio in these deep-water environments

actually is and whether there have been monitoring activities (underwater video) in the fisheries sector to establish the extent of sediment reworking there. From the data on composition of OC in the depocenters, de Haas (1997; thesis) concludes that this OC is very recalcitrant – will it be further mineralised at all? Bakker and Helder (1993) showed that oxygen fluxes in the Skagerrak were not related to the total organic carbon (TOC) content of the sediments, and that apparently only fresh organic carbon from sinking primary production was mineralised. This suggests that the old carbon derived from lateral transport across the large submarine catchment is unreactive. An interesting ancillary question is, whether (and if so, how much) a decline in (postulated) remineralision from artificial disturbance will negatively feed back on primary production and thus will reduce the amount of carbon supplied for burial.

Reply: A spatial comparison of OC densities and the swept area ratio (or other metrics relating to bottom-contact fishing) would indeed be of interest but certainly beyond the scope of this study. We expect that such an analysis would have to include more than simply overlaying OC stocks with swept area ratios. Besides, there is the complication that current OC stocks most likely have already been affected by bottom contact fishing, which complicates the analysis. Our study was also not designed to answer questions on the reactivity of the deposited OC, but we acknowledge that such questions would be relevant in the context of "carbon protection zones".

Action: We briefly discuss reactivity of OC in the Norwegian Trough at the end of section 6.3 now:

Although more research is needed, it is becoming clearer now that seabed disturbance by demersal fishing leads to increased OC mineralisation in cohesive sediments in the short-term (van de Velde et al., 2018) and a general impoverishment in OC in the long-term (Martín et al., 2014). Protecting regional hotspots of OC accumulation from fishing-induced disturbance might therefore be a suitable measure to increase the climate mitigation potential of the seabed. Likely sites that might benefit from protection are to be found in the burial zone (i.e. the Norwegian Trough), while it is unlikely that the turnover zone yields any potential areas worth protecting in this context. Our results could be used jointly with maps showing the footprint of demersal fishing (Eigaard et al., 2016) and other resources to identify potential sites for the establishment of "carbon protection zones". Such management measures that limit the impacted surface area, allowing carbon stocks and faunal communities in the sediment to recover from a disturbance, and resulting in the recovery of carbon burial, might be preferable over technical modifications that reduce the penetration depth of fishing gear (De Borger et al., 2020). *However, such analyses must consider that the OC stocks, as mapped in this study, likely have been affected already by decades of demersal fishing. Our maps therefore do not represent a baseline in a sense of an undisturbed state.*

*Additionally, more research on the reactivity of OC is required to better understand the relationships between OC mineralisation and seabed disturbance. The mineralisation of predominantly refractory OC caused by demersal fishing might be limited or even negligible. In the Skagerrak, oxygen microprofile measurements indicated that mineralisation rates were independent of OC content, but related to the input of fresh OC by primary production (Bakker and Helder, 1993). This suggests that preferentially fresh labile OC was mineralised, while allochthonous OC that accounts for 90% of the OC in the Norwegian Trough (de Haas and van Weering, 1997) might be largely unreactive. Conversely, van de Velde et al. (2018) suggested that OC mineralisation is stimulated after sediment disturbance, likely due to the enhanced decomposition of previously buried refractory OC when it comes into contact with labile OC, a process known as priming* (Steen et al., 2016)*. Another question of interest is to what extent a potential reduction in mineralisation rates due to areal protection of OC stocks might influence primary production and thus supply of OC to the seabed.*

**75: this is only a temporary uptake**

Reply: Agreed, but recovery of benthic species due to spatial protection will likely increase the biomass. If protection ensures higher levels of biomass (compared to the impacted state) in the long-term, this will contribute to carbon drawdown.

Action: The sentence was slightly altered to promote clarity:

Establishment of MPAs protecting against demersal fishing could not only facilitate the recovery of benthic species but also promote *longer-term* carbon uptake by seabed ecosystems *through increased biomass*, as well as prevent further loss of OC stored in sediments (Roberts et al., 2017).

**106: erosion is an unquantified term and there is reason to believe that a substantial fraction of exhumed OC accumulates in recent depocenters**

Reply: It might indeed be preferable to view erosion as the inverse of sedimentation and predict erosion/sedimentation as one variable. However, there is even less information on erosion rates, which renders this approach as currently unfeasible.

Action: No action taken.

**122: which grain density was assumed?**

Reply: 2650 kg m$^{-3}$.

Action: The text was updated to include this information.

**136: water depth = distance to shore? Sedimentation rate according to Müller and Suess is relevant only for deep-sea sediments and vertical transport**

Reply: There is a difference between water depth, which is the vertical distance between the seabed and sea level and distance to shore, which is the horizontal distance between a location and the nearest shoreline. Sedimentation rates might be more relevant in deep sea environments. However, at this stage of the 'conceptual' model building it might be wise to include more rather than less variables that might potentially be relevant. Unimportant variables are subsequently identified through an additional step with the Boruta variable selection wrapper (section 4.1).

Action: The first paragraph of section 3.2 was updated:

The initial selection of environmental predictor variables was based on availability and *potential* relevance. *At this initial stage of conceptual model building* (Guisan and Zimmermann, 2000)*, it might be prudent to include a wide range of potentially relevant variables. A selection of variables that are actually relevant for the model will be performed subsequently.*

**140 ff: What use are geomorphology features beyond water depth and orbital velocities? Why not use numerical values for grain sizes (Bockelmann et al., 2017)?**

Reply: As per previous reply, it is advisable to "cast the net widely" initially. Regarding grain-size data, we included data on sediment fractions (gravel, sand, and mud content) as mentioned in Table 1.

Action: None.

**145: There are no dated cores in several of these areas (judging from a comparison of Figures 1 and 2)**

Reply: For clarity, this sentence was reworded (also to address a comment by reviewer 2).

Action: The sentence reads now:

*These potential accumulation areas were critically reviewed in the light of measured sedimentation rates and geological interpretation of sediment cores (de Haas et al., 1997 and references therein).*

**148: LSR is 0 in most of the area, and besides the oxygen exposure time appears to be not relevant.**

Reply: As mentioned before, it might be preferable to include more potential predictors initially.

Action: None.

**171: Which are these? Include information of statistical weights/relevance as predictor in Table 1**

Reply: Information on selected predictors was 'hidden' in the Supplements. This has now brought into the main document.

Action: Information on selected predictor variables has been added to the results section.

**216 ff: Are these the environmental variables explaining most of the variance? O2 penetration depth was estimated by mud content - is that independent of OC density? Table 1 lists many predictor variables, most of which apparently are not crucial. On the other hand, sedimentation rate is a crucial parameters, but how are grain size/Folk parameters etc. linked to sedimentation rate?**

Reply: No, as per previous comment it should now be clearer which environmental predictors were selected. The six environmental variables mentioned in section 4.4, which were used as input for the regionalisation, were selected based on their expected impact on OC processing.

Action: None.

**221: Show loadings in a figure or table**

Reply: The R Notebook script was updated to provide loadings. However, we would not deem this information so important as to provide it in the main document.

Action: Updated Supplement S4 with loadings.

**243: The mode of transport by OC spiraling is important here - after what time does the material from production sites arrive at the depocenter, and how is the material reworked on the way?**

Reply: Agreed, but we suggest discussing this in section 6.2, which seems better placed than in the results section. The reviewer already raised the question about the origin of the OC in the Norwegian Trough as a general comment and we have provided additional information.

Action: See above.

**247: Are these parameters the dominant discriminators?**

Reply: No, if the reviewer is referring to the selected predictors. As outlined in section 4.4., these parameters were chosen as input data for the regionalisation due to their expected strong impact on OC processing.

Action: None.

**291 ff: This raises the interesting question on who reports -producer or storage provider?**

Reply: Agreed, although this might be beyond the scope of this study.

Action: We added a sentence and refer to a recent publication that deals with this issue.

*This would, however, require new accounting guidance and governance frameworks, as carbon removal from the atmosphere and OC accumulation in seabed sediments might occur in different jurisdictions, with the North Sea cited as a prime example* (Luisetti et al., 2020)*.

**297: The difference is that OC in fjord sediments probably is autochthonous (or from land), whereas Norwegian Trough collects OC from a large submarine area**

Reply: Faust and Knies (2019) show that the proportion of marine vs terrestrial OC varies greatly between fjords. We therefore think that such a generalisation should not be made.

Action: None.

**312: I assume that these characteristic parameters are the ones shown in Figure 6 (bathymetry, oxygen penetration depth derived from mud content, energy at the sea floor from M2 and orbital velocity)? See comments above.**

Reply: Yes, these are the parameters shown in Figure 6, as the regionalisation was based on these.

Action: We added a reference to Fig. 6 to the first sentence of section 6.2.

The regionalisation based on selected characteristic parameters pertaining to OC accumulation *(Figure 6)* and storage has shown that the North Sea and Skagerrak can be divided into distinct zones.

**393: There is always room for improving the data base and suggestions are certainly valid. But when most samples are sands with <0.2 wt.% OC, a standardisation of methods (which effectively has been**

done in the past on various occasions for the analytical steps) will not change the general conclusions. In the de Haas thesis available on the internet (Mededelingen van de Faculteit Aardwetenschappen Universiteit Utrecht No. 155: Transport, preservation and accumulation of organic carbon in the North Sea), there is information on the delta 13C of OC and C/N ratios.

Reply: We agree that a standardisation of methods has been done in the past, otherwise this study would not have been possible. However, such standardisations were probably not being made with OC budgeting in mind. For example, OC contents have been measured at sediment slices of various sizes, while in terrestrial soil mapping standard depths are used (Hengl et al., 2014). The de Haas thesis does indeed include $\delta^{13}C$ and C/N values, however not in a format that could be utilised (i.e. a table).

Action: None.

**395: Again - who is credited for sequestration of allochthonous carbon in the Norwegian trench? Those who produce the carbon (in their EEZ's), or those who store it in their EEZ's? Productivity is highest in the southern, non-depositional sectors of the North Sea. The best way to enhance sequestration potential is to increase productivity – probably by enhancing eutrophication.**

Reply: This point was already addressed twice previously, in the introduction and the discussion. We believe that this should suffice, given that this aspect, despite its future relevance, is not the main topic of this study (as mentioned by the reviewer).

Action: None.

Table 1: Add a measure of statistical weight

Reply: We assume the reviewer is referring to the variable importance measures. This could be done, but we prefer adding this information to the results section.

Action: We now include a figure showing the selected predictors and their importance scores.

Table 2: Reference needed in the caption

Reply: Agreed.

Action: Reference added (Hengl et al., 2017)

Fig. 1: The upper bound of the color scale (51 m) appears to wrong. How were depositional areas selected? What is the diagonal white line from Denmark to the southern tip of Norway?

Reply: Although it might seem wrong, the upper bound is correct. This is due to the relatively large pixel size of the bathymetry grid (500 m). Some 'coastal' pixels will include terrestrial areas and the averaging will have led to positive values. We explain in the text how the depositional areas were derived. The white line between Norway and Denmark constitutes the boundary between the North Sea and the Skagerrak.

Action: If preferable, we could 'force' the upper bound of the colour scale to 0 and remove the white line. Additional text was added to the caption to point to the text where the delineation of depositional areas is explained:

*Refer to chapter 3.2 for the delineation of areas of sediment deposition.*

Fig. 6: Are the parameters presented here as box-and-whisker plots the ones that are dominant, or why were they chosen (see comment above)?

Reply: No. This has been explained previously.

Action: None.

References

[revised manuscript text omitted]

---

## Author Comment (AC2) · 9 Dec 2020

Reply to RC2

Reviewer 2 attests that "this study provides novelty with its spatial approach to mapping OCAR (with associated uncertainty) and is a welcome contribution to net sediment accumulation and depocentre research."

General comments:

Comment: […] the title of the manuscript could be updated to include OCAR for clarity.

Reply: Agreed.

Action: New title reads:

*Organic carbon densities and accumulation rates in surface sediments of the North Sea and Skagerrak*

Comment: The break-down of regions into different 'carbon-processing zones' is a novel concept in mapping, however more thought is required in the discussion as to the physical and biogeochemical processes that transport and cycle carbon on the shelf – and acknowledging the differences between OC accumulation and OC burial (e.g. Zonneveld et al., 2009).

Reply: Following comments from reviewer 1 (RC1), we have already made changes to section 6.2. These relate to the role of bioturbation in OC cycling and the transport of OC to the burial zone. We agree that it is important to stress the difference between OC accumulation and burial and concede that this was not clear enough in the original manuscript. We therefore have added a short paragraph to the introduction that explains the difference between the two. We also rename the burial zone to accumulation zone.

Action: We have inserted a paragraph to the introduction that outlines the differences between OC accumulation and burial:

*It is important to stress the difference between OC burial and OC accumulation here. Burial is the deposition of OC below the zone of active degradation* (Keil, 2015)*. OC degradation in surficial seafloor sediments happens via various processes including aerobic respiration, denitrification, manganese reduction, iron reduction, sulfate reduction and methanogenesis* (Berner, 1980)*. Burial thus is the removal of OC from the active carbon cycle and the burial rate can be expressed as the product of sediment accumulation and OC content at the depth below which no further degradation of OC occurs* (Middelburg, 2019)*. It is, however, difficult to determine that depth. Various depth horizons have been used, e.g. the lower boundary of the sulfate reduction zone* (Jørgensen et al., 1990)*, 15 cm* (Hartnett et al., 1998) *and 10 cm* (Bakker and Helder, 1993)*. OC accumulation rates, on the other hand, can be calculated for any specific depth interval of the sediment column. Due to the difficulties of determining the relevant depth to estimate burial rates and the scarcity of burial rate data, we decided to estimate OC accumulation rates instead.*

Furthermore, we rename the burial zone to accumulation zone and add the following sentence:

This zone might be termed burial zone according to Huettel and Rusch (2000). *However, for consistency with our analysis we term this zone accumulation zone.*

Comment: An additional table of the outputs used to make final OC stock estimates would be useful (for those who can't/don't use R).

Reply: We are not entirely sure what the reviewer is referring to. The calculation of OC stocks is given in section 4.2, equation 6. This is a simple multiplication of the sum of all OC density pixels with the reference depth (0.1 m) and the size of one pixel (250,000 m$^2$). We cannot see the need for a table.

Action: No action taken.

Comment: I question the usefulness of comparing these OC stocks to other stocks; for instance, coastal ecosystems have different mechanisms for sequestering carbon and are spatially limited. Soils are more comparable by area, however presumably there are much more data available due to ease of sampling and therefore lower uncertainties?

Reply: Despite the mentioned differences, we believe that the presented comparisons are useful. We see them as an integral part of our study as they put the shelf sediment OC stocks into context. Soil OC stocks have long been recognised for their carbon storage ecosystem function and so have more recently Blue Carbon ecosystems. The comparisons highlight the importance of shelf sediment OC stocks.

Action: We have added a sentence that explains why the comparisons were made:

*To gauge the importance of North Sea shelf sediments as an OC store, we compare them with coastal habitats and terrestrial soils in the following:*

Comment: Uncertainty estimates in this paper for the sedimentation and OCAR are quite high (same order of magnitude) generally, and I wondered why they were highest in areas with a higher density of data points? (This could be a misunderstanding on my part of the model, but detail would promote clarity!)

Reply: Uncertainties are shown as absolute values, i.e. they have the same unit as the predicted variable. A very different picture emerges when uncertainties are shown as relative values (% of predicted value). This information is currently only displayed in the supplements S1 and S3. If it is deemed important to move the figures into the main document, this could be done.

Action: Currently none.

Comment: I think this paper needs to acknowledge the differences between accumulation rates and burial rates – for instance in section 6.3, it is stated that "zones of OC burial" have been identified, however there was no investigation into how the OC density varied with depth to comment on how effective this site is for burial, and this is an important distinction to make.

Reply: We agree, and as stated before we are now explaining the difference between OC accumulation and burial. Furthermore, we have renamed the burial zone to accumulation zone and any further reference to burial, other than in a general way (e.g. in the introduction), has been removed.

Action: See above.

Comment: Data for the model are limited, with few to no datapoints over large areas of the North Sea and large assumptions are made. Further details would be welcomed relating to data selection, model outputs, interpretation of RSME and variance in the results, and some assumptions could be strengthen by links to the literature (e.g. OC change with depth; oxygen penetration as a function of mud). There is noticeably little discussion of the effect of sediment type on OC which has been shown to be a significant predictor of OC.

Reply: This appears to be a general comment, with more detail given in the specific comments below.

Action: Detailed below in responses to specific comments.

Specific comments:

**Introduction**

It sets the scene well but more clarity needs to be give as to how this study is novel compared to other predictive spatial models for OC stocks (e.g. is it due to a new framework, a different location being studies, or is it about calculating accumulation rates?). The link between OC and sediment type isn't clear, although a focus is made of cohesive sediments. Can the authors expand on what these are and why are the more relevant to OC? Some more detail could be included about the benefits of random forest modelling as a rationale for why this method 'appears' to have been chosen in recent modelling studies. The text from lines 48 – 57 could be strengthened. Why should marine carbon stocks be accounted for and what kinds of damage are possible as a result of disturbance.

Reply: We would argue that the main purpose of the introduction is to set the scene. Aspects about the novelty of the study should be pointed out in the discussion, and apparently this was at least partly successful as the reviewer states that the "break-down of regions into different 'carbon-processing zones' is a novel concept". We did not intend to focus on cohesive sediments specifically or sediment type in general. We would argue that links between sediment type or grain-size and OC content are well-known and would not require specific mention. The section in question was rather intended to briefly summarise the knowledge on demersal fishing impacts on OC in sediments. We do not mention random forest in the introduction, rather, more generically, we refer to machine learning approaches. We have slightly updated the sentence in question to point out advantages of machine learning over geostatistics. As per specific comments below, the text from lines 48 – 57 was strengthened. We added detail on the relevance of stocks and accumulation rates.

Action: Sentence on machine learning and geostatistics slightly altered:

Recent studies appear to prefer machine-learning over geostatistical approaches (Seiter et al., 2004) *due to their performance, flexibility, and generality* (Hengl et al., 2018).

Action: Information on relevance of OC stocks and accumulation rates in the context of management added:

Well-constrained estimates of OC stocks and accumulation rates are also required from a marine management point of view. *OC stocks are a measure of vulnerability potential, while accumulation rates are a measure of the mitigation potential* (Jennerjahn, 2020). The potential of so-called Blue Carbon ecosystems (mangroves, salt marshes, seagrass meadows and potentially macroalgae) to sequester and store OC is an important ecosystem service that has been highlighted in recent years (Duarte et al., 2005; Mcleod et al., 2011; Nellemann et al., 2009). More recently, it has been shown that fjord (Smeaton et al., 2016, 2017) and continental shelf sediments (Diesing et al., 2017) harbour

considerable amounts of OC. In the United Kingdom, the shelf sediment stock (205 Tg) accounts for 93% of OC stored in coastal and marine habitats (Luisetti et al., 2019) and outweighs combined seagrass and saltmarsh stocks (13.4 Tg) by a factor of »15. In Namibia, the marine sediment OC stock is estimated to be larger than the soil OC stock (Avelar et al., 2017). Determining national carbon stocks is essential to understand the potential vulnerability of those stocks to human activities; however, national assessments for greenhouse gas reporting do not account for marine stocks such as organic carbon stored in shelf sediments (Avelar et al., 2017). In Norway, the government has underlined the significance of OC uptake by marine vegetation but OC accumulation in marine sediments is currently not considered (Anon, 2013). […]

**Line 27 – Can the authors suggest what other differences might account for large differences in global stock estimates?**

Reply: We believe that the estimates of Atwood et al. (2020) are far too high due to the assumptions that have been made in that study. Specifically, they standardised to a depth of 1 m by taking the average OC stock per centimetre and multiplying it by 100, i.e., they did not account for reductions in OC with increasing depth. Other assumptions might also be questionable, e.g., the application of a pedotransfer function that shows very high scatter in the data range typical for OC content of marine sediments. However, we felt it would be distracting to discuss these issues in the introduction. It would also be difficult to sum the above up in a simple, short sentence; hence we decided to not address this in the discussion.

Action: No action taken.

**Line 39 – Misleading - suggests the authors will look at burial rates as well as accumulation.**

Reply: Agreed.

Action: We have now inserted a paragraph detailing the differences between the two and mention that our study only deals with OC accumulation.

**Line 44 – Suggest the authors make reference to these fjord studies coming from the UK (other fjordic studies are available if the authors wanted a more global perspective in this argument). Is this study trying to improve the North Sea estimates specifically or estimates generally? (Line 41).**

Reply: The main statement of the sentence in question is that beyond Blue Carbon ecosystems, fjordic and open marine sediments harbour large amounts of OC. We refer to some foundational studies which attempted to estimate such OC stocks. We are unsure whether the reviewer thinks we should have referenced additional studies from the UK?

Action: No action taken.

**Line 47 – The inclusion of Namibia is unexpected in this comparison. What is the relevance?**

Reply: Generally, it is assumed that terrestrial soil OC stocks are larger than marine counterparts. In Namibia, this appears not to be the case, presumably due to upwelling offshore (high OC) and desert environments onshore (low OC). The sentence was included to highlight this specific situation.

However, we would be willing to delete the sentence if the reviewer thinks it is confusing. The other reviewers seem to not think that way.

Action: Sentence will be deleted if required.

**Line 53 – Suggest removing 'projects' (replace with strategy?) – I don't think stocks themselves can be used to mitigate against GG emissions.**

Reply: Agreed.

Action: Replace projects with strategies.

**Line 55 – Can the authors provide some detail about the kinds of damage attributed to carbon release?**

Reply: Agreed.

Action: The types of damages have been specified in brackets for coastal (areal loss of seagrass habitats, sediment OC loss from saltmarshes) and shelf sea sediment (resuspension by bottom contact fishing).

**Line 59 – To strengthen the idea of using MPAs, can the authors provide some detail as to how MPAs have been used to protect BC carbon storage?**

Reply: Agreed.

Action: We have added "by slowing, halting, or reversing the trend of degradation and loss of e.g. seagrass and mangrove ecosystems. In Indonesia, MPAs reduced mangrove loss by about 140 $km^2$ and avoided emissions of 13 Tg $CO_2$ equivalent between 2000 and 2010 (Miteva et al., 2015)."

**Line 74 – A note to reflect on the likelihood of MPAs (especially on this scale) being developed to protect the seabed against demersal fishing – this isn't a straight-forward decision.**

Reply: We agree, this will be a contentious issue, but big challenges require bold solutions. However, this is beyond the topic of the paper.

Action: No action taken.

**Line 79 – It is not clear how linking to an area most heavily impacted by human activities is ideal for understanding accumulation rates – the study isn't necessarily looking at the effects of human activities on accumulation rates.**

Reply: Maybe not, but this study aims at estimating OC stocks and accumulation rates in a marine environment that is impacted by human activities, hence the discussion of potential impacts of bottom contact fishing on OC stored in surficial seafloor sediments.

Action: We have changed the sentence which now reads: "This makes the area ideal for our study, which has the objectives to estimate OC stocks and accumulation rates of surface sediments in a regional sea that is impacted by human activities."

**Data**

Generally, some more detail is requested for the final datasets used (there are large areas of the North sea with no data – do they not exist?), some of the assumptions made on sedimentation rates and the criteria used to assign accumulative areas. Are figures or supplementary datasets available for the oxygen penetration depth and oxygen exposure time? I'm not clear from the text what form these data take – continuous raster layers? Oxygen exposure time is calculated using the sedimentation rate which is modelled within this study – so the uncertainties will be carried across presumably. Are the Haas data reliable? Some more detail on why certain values were changed and the criteria used to make these decisions would be useful.

Reply: Additional information on the datasets, assumptions on sedimentation rates and criteria used to assign accumulative areas is given, see replies to specific comments below. Figures showing all predictor variables are now provided as well. We also provide additional detail on the deselection of four sedimentation rate values (see below).

**Line 108 – Can the authors elaborate on what pseudo-observations are and if they are comparable?**

Reply: Agreed.

Action: Section 3.1.1 was amended to give a more detailed explanation on pseudo-observations:

*Pseudo-observations are 'virtual' samples that are placed in undersampled areas and for which the value of the response variable can be assumed with high certainty. Hengl et al. (2017) cite 0 % soil OC in the top 2 m of active sand dunes as an example. The placement of pseudo-observations was restricted to* areas of erosion and non-deposition (based on the sedimentary environment layer, as described in chapter 3.2)*, for which* a sedimentation rate of 0 cm yr$^{-1}$ could be assumed. *The pseudo-observations were placed randomly to avoid human bias.*

**Line 111 – What was it about the 210-Pb profiles that made the authors reject some data?**

Reply: Sentence was changed to give additional information.

Action: The following information was added:

*…due to low $^{210}$Pb activities and indistinct decreases with depth.*

**Line 116 – Suggest making a reference to Supp Data Table and provide some more detail in the text for these data. Where have the OC measurements come from? How many etc.**

Reply: Agreed.

Action: The first part of section 3.1.2 was rewritten:

Previous studies have predicted OC concentrations and sediment porosity separately to calculate OC stocks (Diesing et al., 2017; Lee et al., 2019; Wilson et al., 2018). Here, we first calculate OC density from concurrent measurements of OC concentrations and sediment dry bulk densities or porosities. This has two advantages: First, there is no need to transform the response variable as would be necessary in the case of OC concentrations reported as weight-% or fractions. Second, only one model instead of two needs to be fitted. *This is advantageous as fitting two models would likely increase the uncertainty of the predictions. Initially, a wide range of data sources were accessed. Ultimately, 373 samples fulfilled the criterion of providing OC content and dry bulk density/porosity measured on the same sample. These samples were collected and measured by the Geological Survey of Norway, the Centre for Environment, Fisheries and Aquaculture Science, Bakker and Helder* (1993) *and de Haas et al.* (de Haas et al., 1997). *The full dataset used for subsequent modelling is shown in Figure 2 and provided as Supplementary Data Table 2.*

**Line 124 – Refer to Supp Data Table for reference.**

Reply: We now refer to the Supplementary Data Table a few lines before, so maybe this is no longer necessary.

Action: No action taken.

**Line 143 – Suggest including a figure to show the Folk classes of the area and the 'cleaned' boundaries.**

Reply: Agreed.

Action: We provide an additional figure as supplementary material.

**Line 145 – What criteria were used to decide whether an area was potentially accumulative or not?**

Reply: The sentence was rewritten to provide more clarity.

Action: The sentence now reads:

*These potential accumulation areas were critically reviewed in the light of measured sedimentation rates and geological interpretation of sediment cores (de Haas et al., 1997 and references therein).*

**Line 147 – Can the authors describe generally what the relationship between measured oxygen depth and mud content is expected to be? Does oxygen penetrate more or less in mud? What is the relationship to cohesive sediments?**

Reply: According to the equations provided in Table 1, oxygen penetration depth decreases with increasing mud content up to 8 weight-%. Further increases in mud content do not affect the oxygen penetration depth.

Action: We now provide maps of all predictors including oxygen penetration depth, as requested by reviewer 1. This might help get a better understanding of how oxygen penetration depth varies spatially.

**Method**

The use of the QRF Random Forest model is well justified, and the methods are clear / concise. Some detail on what the different types of error / variance generated mean would be useful and how this differs from the coefficient of determination.

Reply: There was an error in the formula for calculating the explained variance. This effectively means that $r^2$ and variance explained are essentially the same.

**Line 198 – Would be useful to provide a conversion factor to OC stocks from other studies referenced in this study e.g. Tg - Mt / Tmol and that use different units. This would make inter-study comparisons easier / more transparent.**

Reply: Agreed.

Action: We now provide additional information regarding the conversion of Tg to Mt and Tmol:

*OC stocks and uncertainties are reported in Tg. One Tg equals 1 Mt or 0.083 Tmol C.*

**Line 204 - Somewhere it should be noted that there is a difference between carbon accumulation rates and burial rates (i.e. just because carbon is accumulating, doesn't automatically mean it is being buried in the same amounts).**

Reply: See reply to general comments. We now provide a paragraph in the introduction that explains differences between OC accumulation and burial. We also state that we estimate OC accumulation.

Action: No further action deemed necessary here.

**Line 221 - Specifically what were these variables that accounted for 95.5% of the variance?**

Reply: Principal components might have contributions from all variables. The loadings are now provided as part of Supplement S4 – Regionalisation. However, we do not show this information in the main document. It would be difficult to express the contributions verbally in a simple manner, and this might distract from the main points. The main message of the sentence in question is that a very large part of the variance of the initial variables can be expressed with four principal components, and it is these which are subsequently used for clustering (regionalisation). We also provide boxplots of the six variables along with the spatial representation of the regions, which allow to infer typical characteristics of the three regions.

Action: Updated Supplement S4 with loadings.

**Results**

Concise reporting – although it is not entirely clear how to interpret / use the RMSE and Explained Variance values. A table showing how the final results have been derived would be useful – can the model output results at specific stages? A breakdown of the average sedimentation / OC density and

OCAR results by the three regions would be useful. It is not clear to me why there is higher uncertainty in higher sedimentation rates which is also where there is a higher density of data points. The results section might not be the correct section to answer this but do the authors have any insights into why there is a much higher proportion of OC accumulating (87%) in the Norwegian Trough than the proportion stored here (25.9%) – Is there high turnover here? The discussion mentions several characteristics of this area which enhance preservation of OC.

Reply: We still struggle to understand "how a table showing how the final results have been derived would be useful". What exactly is meant with final results? If it is the OC stock, then calculations were made as outlined in equation 6 based on the predicted pixel values, the size of a pixel and the reference depth. We also provide information on OC density and accumulation rates per region in Figure 6. Regarding higher uncertainty in areas of higher sedimentation rates despite more data points: Figure 3 shows absolute uncertainty (same unit as predicted variable). Absolute uncertainties would be expected to increase with increases in predicted values. However, the relative uncertainty (% of predicted value) gives a very different picture. This information is currently only displayed in the supplement S1. If it is deemed important to move the figures into the main document, this could be done. Regarding higher proportions of OC accumulating then being stored in the Norwegian Trough: This might be due to the somewhat arbitrary reference depth of 10 cm that was chosen. If we would be able to choose a reference age (with variable depth across the basin), this would probably account for the differences mentioned. Unfortunately, this is (currently) not possible.

**Discussion**

 #Relevance – This section can be strengthened. Perhaps the section needs to be re-titled to "Context". There are many assumptions made (for instance how OC changes with depth), which increase the uncertainty in the scaled-up estimates (making it less useful for improved carbon stock accounts). The discussion on reporting uncertainties could reflect on how to improve uncertainties. The authors argue that their uncertainty estimates are robust because they are based on soil OC mapping studies which, will be different to the marine realm because sampling is easier and there are different predictor variables influencing OC distributions presumably. The comparison of shelf sediment stocks to coastal "blue carbon" doesn't acknowledge the differences between the ecosystems e.g. that coastal habitats are spatially limited to the intertidal zone, have a much smaller areal coverage and has a different mechanism in terms of carbon sequestration. The argument for the Norwegian Trough as a unique and highly effective zone of carbon accumulation (if this is what the authors are trying to argue) needs to be re-worked for emphasis – it gets lost by the introduction of Scottish and Irish fjords.

Reply: 'Context' might be an equally appropriate title of the section, but we called it 'Relevance' because it was intended to outline the relevance of North Sea OC stocks and accumulation in a wider context. Globally, soils are named as an important OC store, so we wanted to compare our results with those stocks. Likewise, Blue Carbon ecosystems and fjords have been named as important places of OC accumulation and we make comparisons in that direction. Several reviewers commented on the upscaling of the OC stocks of Lee et al. (2019) from 5 cm to 10 cm depth. Apparently, this section missed clarity, so we removed it, as LaRowe et al. (2020) provide the estimate we require (i.e., top 10 cm of global shelf sediments). We briefly discuss in the beginning of section 6.4 that the most likely source for high uncertainties are the available samples due to biases regarding the coverage of the temporal, geographical and predictor variable space. We then go on to explain that the provided uncertainty maps could be used to guide additional sampling with the aim

to reduce uncertainty. We did not argue that our uncertainty estimates are robust because they are based on soil OC mapping studies. Rather, we argue that they are robust because they consider two sources of uncertainty: uncertainty in the model and in variations of available data. To do so, we adapted a methodology that was developed for soil OC mapping. We now briefly acknowledge that Blue Carbon ecosystems differ in terms of areal coverage and carbon sequestration. The argument for the Norwegian Trough as an effective zone of OC accumulation has been emphasised.

Action: The first paragraph of section 6.1 reads now:

*The surface sediments of the North Sea and Skagerrak store 230.5 ± 134.5 Tg of OC. This compares with 9.6 to 25.0 Pg C stored globally in bioturbated Holocene shelf sediments (0 – 10 cm) as estimated by LaRowe et al.* (2020)*. Hence, sediments in the North Sea and Skagerrak store approximately 0.9 – 2.4 % of the global stock in an area that accounts for ≈1.7 % of the global shelf.*

**Line 260 - A figure would be useful to put the 'global continental shelf' in the context of the global seafloor (and then the two regional seas into context as well).**

Reply: This sentence has been removed, hence no longer relevant.

Action: None.

**Line 264 – The assumption is very vague - are there any studies that provide an estimate of how OC stock changes with depth to get a narrower estimate?**

Reply: This sentence has been removed, hence no longer relevant.

Action: None.

**Line 270 - Where does 58% OC stock uncertainty come from? Line 233? (Explained variance?)**

Reply: Uncertainty divided by estimate times 100.

Action: The text was updated:

*When comparing uncertainties in OC stock estimates with other reported values of spatial predictions at a regional to global scale, we find that our value of 58 % (100 * 134.5 Tg / 230.5 Tg) is similar to that reported by Lee et al.* (2019) *amounting to 49 %, while other studies did not report any estimates of uncertainty* (Diesing et al., 2017; LaRowe et al., 2020)*.*

**Line 272 – The comparison to lower uncertainty values from local studies could be further developed.**

Reply: Agreed.

Action: Sentences added:

*An intrinsic assumption of modelling approaches such as the one presented here is that the measured response variable is representative at the scale of the pixel size of predictor variables. The likelihood for this being true increases when the pixel size approaches the size of the seabed area that was*

*sampled with a grab or corer. Higher resolution predictor variables, as frequently used in local studies, might therefore have lower uncertainties associated with the predictions.*

**Line 274 - Is this a good comparison? Does soil OM have similar predictor variables (e.g. current speed?) Soils are presumably easier to sample as well and therefore have a better spatial range of samples. Some further development of the argument would be helpful.**

Reply: Terrestrial counterparts of marine sciences are generally years if not decades ahead regarding methodology. We think it is therefore prudent to transfer developed methodologies from the terrestrial to the marine realm. The mentioned methodology for soil OC mapping is sufficiently generic that we cannot see how a transfer of such knowledge would be detrimental. However, our wording might be confusing, and we therefore rephrased the sentence.

Action: The sentence now reads:

*We believe that our approach to uncertainty assessment is very robust as it estimates uncertainty in the model and in variations of available data.*

**Line 282 - Coastal habitats are limited spatially by depth and limited to coastlines - generally intertidal zone which is not considered the continental shelf. Can the authors provide an area estimate for these coastal habitats to provide context for the OC-stock values reported? How do the OC densities compare when normalised to area?**

Reply: Agreed.

Action: We provide additional information on coastal habitats:

*Coastal vegetated habitats (saltmarsh, seagrass, kelp and tidal flat) are known to bury large amounts of carbon despite occupying only 0.2 % of the global ocean surface (Duarte et al., 2005, 2013). Coastal habitats  on the northwest European continental shelf store between 8.3 and 40.8 Tg C in the upper 10 cm in an area of 20,900 – 35,000 km$^2$ (Legge et al., 2020), equating to OC densities between 24 and 195 kg m$^{-3}$. This indicates that shelf sediment stocks (230.5 Tg) are approximately an order of magnitude larger despite lower OC densities of 1.1 to 13.6 kg m$^{-3}$.*

**Line 293 - The word project is ambiguous and implies that sediments can be managed to increase sequestration of CO2. The link between greenhouse gases and OC found in sediments is not made. What are the implications for accounting for these stocks? National inventory numbers would increase - but how can this be useful of greenhouse gas reporting?**

Reply: After re-reading the paragraph, we felt that the whole sentence might be slightly misplaced. We therefore decided to discuss such issues in section 6.3 (Implications for management).

Action: A new paragraph has been added to section 6.3 after the first paragraph:

*Marine sediment OC stocks are presently not considered in the context of national carbon inventories for greenhouse gas reporting. The question has been raised whether those stocks should be considered as part of national carbon accounting (Avelar et al., 2017). It is becoming clearer that marine sediments store sizeable amounts of OC (Diesing et al., 2017; Lee et al., 2019; Luisetti et al.,*

2019)*, which might be vulnerable to human activities such as demersal fishing. Likewise, there exist hot spots of OC accumulation* (Bianchi et al., 2018) *like the Norwegian Trough, as demonstrated here. A further exploration as to how management of marine sediment OC could contribute towards national greenhouse gas emission reduction targets might therefore be prudent; however, this requires new accounting guidance and governance frameworks* (Luisetti et al., 2020)*. The assessment of the OC stock size should be coupled with an assessment of the anthropogenic impacts on that stock* (Avelar et al., 2017)*. When assessed in the context of naturally occurring disturbance (e.g., by currents and waves), this will contribute towards a more complete picture of the vulnerability of marine sediment OC stocks to remineralisation and potential release of $CO_2$ to the atmosphere* (Atwood et al., 2020)*. We provide spatially explicit information on stock sizes and the uncertainty in the estimates, which could be utilised in such vulnerability assessments.*

**Lines 300 – 304 - Suggest rearranging - I think the authors are trying to say that the Norwegian trough could be an OC accumulation zone unique even to fjord environments because it is apparently not heterogeneous?**

Reply: Agreed.

Action: We have rearranged the paragraph, which now reads:

*The accumulation of OC is effectively limited to the Norwegian Trough, with the highest rates found in the Skagerrak. Predicted OCARs vary between approximately 4 and 66 g $m^{-2}$ $yr^{-1}$ in the Norwegian Trough, with a mean OCAR of 19.4 g $m^{-2}$ $yr^{-1}$. Reported OCARs measured in fjord sediments in Norway and Sweden bordering on the North Sea and Skagerrak range from 12 to 54 g $m^{-2}$ $yr^{-1}$* (Huguet et al., 2007; Müller, 2001; Nordberg et al., 2001, 2009; Skei, 1983; Smittenberg et al., 2004, 2005; Velinsky and Fogel, 1999)*, indicating that OCARs in the Norwegian Trough are of a comparable magnitude. However, fjords in Scotland and Ireland have been shown to be heterogeneous in sediment distribution and OC concentrations* (Smeaton and Austin, 2019)*, and hence also OC accumulation. Judging from published sediment maps* (e.g. Elvenes et al., 2019)*, the same applies to fjords in Norway. Conversely, the Norwegian Trough is characterised by fine-grained sediments* (Mitchell et al., 2019) *and OC accumulation occurs throughout the geomorphological structure. Additionally, the area of the Norwegian Trough is much larger than even the largest fjords in Norway, highlighting its relevance as the most important place of OC accumulation in the North Sea and Skagerrak.*

**Line 303 – Reference / figure to back-up that the Norwegian trough has homogenous sediment?**

Reply: Agreed.

Action: Reference added (see above).

**Zones of OC processing at the seafloor – The first paragraph is too reflective and needs a few more references for statements. It isn't clear how initial studies of OC cycling on the shelf led to the notion of rapidly accumulating coastal sediments? The authors provide a useful summary of environmental seafloor processes to explain oxygen dynamics.**

Reply: We are not sure what the reviewer means with "too reflective". Is this comment referring to the first two sentences, which summarise our results regarding the regionalisation? We would deem

this rather an appropriate way of introducing the discussion on the zones of OC processing. Initial process studies focussed on accumulation areas and this might have led to the notion of rapidly accumulating coastal sediments. References were given for this.

Action: No action taken.

Lines 339-342 are unclear that the characteristics listed are for sediment properties that influence OC cycling. Needs a little re-working.

Reply: We would have thought that the reasoning provided is clear, as we state frequently cited factors that increase the potential for OC preservation. However, as reviewer 3 also requests rewording to promote clarity, we provide additional detail.

Action: The sentences were changed:

*This lack of advective oxidation* (Huettel et al., 2014; Huettel and Rusch, 2000) *translates into slower OC degradation. Fine-grained sediments provide mineral protection* (Hedges and Keil, 1995; Hemingway et al., 2019; Keil and Hedges, 1993; Mayer, 1994)*, which also promotes OC preservation. Short oxygen exposure times* (Hartnett et al., 1998) *due to shallow oxygen penetration depths and relatively high sedimentation rates limit the time for aerobic mineralisation. Collectively, this leads to high OC densities and accumulation rates.*

**Implications for management – This section is currently too vague. Although the implications of refining zones due to OC processing is an interesting concept and potentially a useful way of simplifying areas for management, the scales discussed for MPAs are probably too large to be effective or manageable. Natural disturbance hasn't been acknowledged.**

Reply: This section has been strengthened by the inclusion of two paragraphs on OC stocks in the context of greenhouse gas emissions accounting (see above) and the reactivity of OC (following a comment from reviewer 1). We do not discuss any scales for MPAs in our text. Rather, we provide information on OC accumulation rates. Based on these data and in conjunction with other pieces of evidence, it might be possible to identify sites for MPAs. A brief mention of natural disturbance has now been made in the newly added text.

**Line 351 – "Potential zones of OC burial" - there was no investigation into how the OC density varied with depth to comment on how effective this site is for burial – this should be removed.**

Reply: As stated earlier, we have tightened the terminology and are now referring to zones of OC accumulation.

Action: No further action.

**Suggestions for Future Research – More detail is required around further data collection – the goal of data collection needs to be elaborated and more thought into specifically what data would be useful / beneficial to collect to answer questions relating to carbon stocks. The sampling design examples are very technical - who might undertake this enormous task? Some detail about existing data stores would be useful for reader understanding that national and pan-European datacentres do**

exist. I think some further discussion on the ideas behind 'source of OC' – and why this might be relevant to further study in terms of thinking about climate mitigation – is needed. The authors presume this is common knowledge.

Reply: We provide more information on parameters to be measured now. However, this section was not meant to be an exhaustive discussion on which parameters would be useful to answer specific research question. It was rather meant to stimulate debate. We would consider our suggestions regarding sampling design as specific, which might in fact be useful. We also mention that such methods help minimising the sampling effort. We now refer to existing databases. We explain in the text that the source of OC (terrestrial or marine) is relevant for carbon offsetting schemes, as such schemes would not allocate offset-credits for allochthonous (i.e., terrestrial) OC due to the risk of duplicating carbon sequestration gains that have already been accounted for in adjacent ecosystems.

**Line 375 – Further discussion about what new samples are being recommended for collection. To collect what specifically? grain size? Carbon measurements? to what depth? In-situ oxygen / current data / sedimentations rates?? What are the questions / gaps to inform what data are required?**

Reply: Additional information is given.

Action: Sentence now reads:

*Consequently, there is a need for the collection and analysis of new samples on OC content, dry bulk density, sedimentation rates and ancillary parameters (e.g., grain size).*

**Line 378 – Could you elaborate on how economic benefits can be achieved?**

Reply: Agreed.

Action: Sentence added:

*Jin et al. (2020) developed an analytical model of the economic effects of global carbon emissions including uncertainty about biological carbon pump sequestration and estimated that the benefit to narrow the range of uncertainty about ocean carbon sequestration is on the order of $ 0.5 trillion.*

**Line 380 – What type of baseline dataset?**

Reply: There is additional information given at the end of section 6.3.

Action: Additional text:

*However, such analyses must consider that the OC stocks, as mapped in this study, likely have been affected already by decades of demersal fishing. Our maps therefore do not represent a baseline in a sense of an undisturbed state.*

**Line 388 – Agreed - however there are national sampling programmes that have standardised protocols – do these need to be advertised' to the research community / or informed by the research community?**

Reply: We agree that a standardisation of methods has been done in the past, otherwise this study would not have been possible. However, such standardisations were probably not being made with OC budgeting in mind. For example, OC contents have been measured at sediment slices of various sizes, while in terrestrial soil mapping standard depths are used (Hengl et al., 2014).

Action: None.

**Line 391 – Such facilities do exist – e.g. ICES.**

Reply: Yes, but ICES is collecting data on a wide range of parameters. We were thinking of a repository which is more specific to sediment carbon. It appears that such a database has just been made public.

Action: Sentence altered:

*Although facilities to store and retrieve quality-controlled seafloor data centrally exist (e.g., EMODnet, ICES), it would still be advantageous to establish global data archives that are more specific to marine sedimentary carbon such as MOSAIC* (van der Voort et al., 2020)*.*

**Line 396 – How likely is there to be terrigenous OM in shelf sediments? Any studies that have looked at this?**

Reply: There is limited information in the PhD thesis of de Haas (1997). However, the paragraph was meant to indicate that more data are required.

Action: None.

Technical Corrections

Comments are provided with specific line references for consideration:

**Line 7 – Suggest re-wording; Sediments don't protect the seabed from disturbance, Sediments can store carbon, provided left undisturbed (from anthropogenic activity).**

Action: Text changed to "e.g., by storing organic carbon if left undisturbed from anthropogenic activity."

**Line 10 – Inclusion of 'us' between 'allows to'.**

Action: Agreed.

**Line 16 – Suggest updating 'on par' with 'comparable'.**

Action: Agreed.

**Line 30 – Suggest replacing 'were' with 'have been'.**

Action: Agreed.

**Line 31 - Suggest replacing 'did not include' with 'have not included'.**

Action: Agreed.

**Line 33 – Reference for importance of continental margins in OC cycling – and important in what way?**

Action: This was detailed in the previous paragraph. No action.

**Line 37 – Use of the word 'appear' without suggesting why this might be. What are the advantages of machine learning over geostatistical approach?**

Action: Text updated:

…*due to their performance, flexibility, and generality* (Hengl et al., 2018)

**Line 41 – Suggest replacing 'point of view' with 'perspective'.**

Action: Agreed.

**Line 42 – Reference for the inclusion of 'potentially macroalgae' in the BC definition.**

Action: Reference (Krause-Jensen and Duarte, 2016) included.

**Line 63 – Add 'The'.**

Action: Agreed.

**Line 64 – Suggestion for consideration. Is 'fertilization' the right term? to fertilise means to stimulate productivity - this would reduce OC presumably. Is enrichment a better term?**

Action: This is the word used in the cited reference. No action.

**Line 66 – Suggest including 'sediment' between 'deeper layers'.**

Action: Agreed.

**Line 69 – Change 'expectable' to 'expected'.**

Action: Agreed.

**Line 78 – suggest replacing 'it is one of the regional seas' with 'they are the' for comprehension.**

Action: Agreed.

Regional setting – Figure 1 - Request to add the labels for the two regional seas on figure 1 location map.

Action: Agreed.

**Line 89 – Suggest re-wording 'generally deepening from south to north'. Specific depths?**

Reply: Is this necessary information that couldn't be gleaned from Figure 1?

Action: None.

**Lines 117, 118, 120 – Use of the word concentration is incorrect. Update to content (mass per unit mass) – See Flemming & Delafontaine, 2000.**

Action: Agreed.

**Line 133 – suggest adding 'relevance to OC'.**

Action: Agreed.

**Line 139 – Suggest addition of appropriate reference to reinforce-up this statement.**

Reply: This would appear to us as common knowledge

Action: None.

**Line 155 – Suggest replacing 'target' with 'response' to keep the terms consistent.**

Action: Agreed.

**Line 158 – Inclusion of the word 'us'.**

Action: Agreed.

**Line 183 -Would be useful to include a sentence describing what the RMSE explains (and the difference between this and the MSE in the context of the model performance)**

Action: Sentence modified:

RMSE *measures how far apart on average predicted values are from observed values. It* might range from 0 to infinity, with an ideal value of 0.

**Line 273 - Suggest replacing 'how' with 'in which'.**

Action: Agreed.

**Line 283 - 284 – Sentence doesn't make sense compare to preceding sentence.**

Action: No longer relevant due to changes made to paragraph.

**Line 285 – Does 'collectively' mean 'global'?**

Action: Changed to globally.

**Line 315 – References to initial process studies?**

Action: References added:

(e.g. Balzer, 1984; Jørgensen, 1977; Martens and Val Klump, 1984)

**Line 316 – How did one lead to the other?**

Reply: What is meant with 'one' and the 'other'? We are simply naming sediment characteristics in cohesive, diffusion-dominated sediments.

Action: None.

**Line 339 – 341 – Sentence isn't well constructed or complete.**

Action: Sentence has been changed (see above).

**Line 342 – Suggest inclusion of 'a' burial zone.**

Action: Agreed.

**Line 350 – Suggest replacing 'on par' with 'comparable'.**

Action: Agreed.

**Line 351 – Suggest replacing 'act differently' with 'have different roles'? ('act' suggests it is a conscious action - not a by-product of location and physical environment**

Action: Agreed.

**Line 354 – I don't understand the point about 'total annual rate in the North Sea'.**

Action: Replaced with 'the annual rate of OC accumulation by coastal vegetated habitats (Legge et al., 2020).

**Line 370 – Suggest adding 'However' at start of sentence.**

Action: Agreed.

**Line 382 - Relative importance on what? I assume OC but this isn't explicit.**

Action: Added "OC distribution".

**Line 405 – Suggest replacing 'on par' with comparable.**

Action: Agreed.

Literature

[revised manuscript text omitted]

---

## Author Comment (AC3) · 9 Dec 2020

Reply to RC3

The reviewer attests that "the work presented in this manuscript represents a valuable contribution to the field that should be published."

General comments:

-

Specific comments:

Lines 108-109: Give a short definition of "pseudo-observations" in the context of this work.

Reply: Agreed.

Action: Text was updated:

*Pseudo-observations are 'virtual' samples that are placed in undersampled areas and for which the value of the response variable can be assumed with high certainty.*

Lines 111-112: Define how many are meant by "Some of the sedimentation rate values: : :", does this refer to the four values that are amended later in the same sentence or are these four a subset of the "some"? If it's a subset, the selection process should be explained.

Reply: Agreed.

Action: Sentence now reads:

Four of the sedimentation rate values from non-depositional areas reported by de Haas et al. (1997) and van Weering et al. (1993) were set to 0 cm yr$^{-1}$ *due to low $^{210}$Pb activities and indistinct decreases with depth*.

Line 131 (Figure 2): There seem to be no OC measurements in the Elbe Paleo valley region (Region 2), if this is the case it should be explicitly mentioned.

Reply: de Haas et al. (1997) cite an average deposition rate of 1 mm yr$^{-1}$ (Eisma, 1981). However, no precise location is given, hence the rate is not included in the dataset on sediment rates.

Action: None.

Line 145: "critically reviewed and removed if they were not deemed accumulative" an explanation on the selection/removal criteria should be added here.

Reply: The sentence was rewritten to provide more clarity.

Action: The sentence now reads:

*These potential accumulation areas were critically reviewed in the light of measured sedimentation rates and geological interpretation of sediment cores (de Haas et al., 1997 and references therein).*

Lines 263-264: "It is therefore safe to assume that the sediment slice between 5 and 10 cm will contain between 0 % and 100 % of the OC stock of the upper 5 cm." It is generally safe to assume that anything contains between 0% and 100% of anything, so this sentence is either unnecessary, or should be reworded in a way that makes more sense.

Reply: The sentence has been removed as the beginning of the section was restructured.

Action: No further action.

Lines 339-341: "Lack of advective oxidation […] and relatively high sedimentation rates." The wording of this sentence is unclear and should be revised.

Reply: Agreed, we provide additional detail.

Action: The sentences were changed:

*This lack of advective oxidation (Huettel et al., 2014; Huettel and Rusch, 2000) translates into slower OC degradation. Fine-grained sediments provide mineral protection (Hedges and Keil, 1995; Hemingway et al., 2019; Keil and Hedges, 1993; Mayer, 1994), which also promotes OC preservation. Short oxygen exposure times (Hartnett et al., 1998) due to shallow oxygen penetration depths and relatively high sedimentation rates limit the time for aerobic mineralisation. Collectively, this leads to high OC densities and accumulation rates.*

References

Eisma, D.: Supply and Deposition of Suspended Matter in the North Sea, Spec. Publ. Int. Ass. Sediment., 5, 415–428, doi:10.1002/9781444303759.ch29, 1981.

de Haas, H., Boer, W. and van Weering, T. C. E.: Recent sedimentation and organic carbon burial in a shelf sea: the North Sea, Mar. Geol., 144, 131–146, doi:10.1016/S0025-3227(97)00082-0, 1997.

Hartnett, H. E., Keil, R. G., Hedges, J. I. and Devol, A. H.: Influence of oxygen exposure time on organic carbon preservation in continental margin sediments, Nature, 391, 572 [online] Available from: http://dx.doi.org/10.1038/35351, 1998.

Hedges, J. I. and Keil, R. G.: Sedimentary organic matter preservation: an assessment and speculative synthesis, Mar. Chem., 49(2–3), 81–115, doi:10.1016/0304-4203(95)00008-F, 1995.

Hemingway, J. D., Rothman, D. H., Grant, K. E., Rosengard, S. Z., Eglinton, T. I., Derry, L. A. and Galy, V. V: Mineral protection regulates long-term global preservation of natural organic carbon, Nature, 570(7760), 228–231, doi:10.1038/s41586-019-1280-6, 2019.

Huettel, M. and Rusch, A.: Transport and degradation of phytoplankton in permeable sediment, Limnol. Oceanogr., 45(3), 534–549, doi:10.4319/lo.2000.45.3.0534, 2000.

Huettel, M., Berg, P. and Kostka, J. E.: Benthic Exchange and Biogeochemical Cycling in Permeable Sediments, Ann. Rev. Mar. Sci., 6(1), 23–51, doi:10.1146/annurev-marine-051413-012706, 2014.

Keil, R. G. and Hedges, J. I.: Sorption of organic matter to mineral surfaces and the preservation of organic matter in coastal marine sediments, Chem. Geol., 107(3–4), 385–388, doi:10.1016/0009-2541(93)90215-5, 1993.

Mayer, L. M.: Surface area control of organic carbon accumulation in continental shelf sediments,

Geochim. Cosmochim. Acta, 58(4), 1271–1284, doi:10.1016/0016-7037(94)90381-6, 1994.

van Weering, T. C. E., Berger, G. W. and Okkels, E.: Sediment transport, resuspension and accumulation rates in the northeastern Skagerrak, Mar. Geol., 111(3), 269–285, doi:https://doi.org/10.1016/0025-3227(93)90135-I, 1993.

---

## Author Comment (AC4) · 9 Dec 2020

Reply to SC1:

Line 87: In the Regional setting or Data sections it would be worthwhile stating why the boundaries of the study site were selected. I presume this was due to the overlapping extent of predictor variables listed in Table 1. However, it is unusual that the focus does not cover the complete extent of any of the countries EEZ presented in this study (Table 2), because had it done so this would improve the impact of the current piece of work. As the discussion encourages further research of this kind (Section 6.4 and the publishing of R scripts), it would be worth clarifying whether similar predictor data are available, or whether these too would need to be generated first. If on the other hand it was due to available sample data or how far the authors felt they could extrapolate the models then this would also be of interest to future scientists doing similar work.

Reply: Different options exist how a study site might be defined. We focussed our work on a sea basin, i.e., the North Sea and Skagerrak as defined by IHO (1953). It was not our aim to present OC stocks and accumulation rates of a specific EEZ or other management unit. It will likely be necessary to gather suitable predictor variable layers for other studies that wish to utilise our code.

Line 190: There appears to be an error in the calculation of VE and r^2. While r^2 is also termed the coefficient of determination, it is my understanding that the VE and r^2 are the same metric. Therefore, I was surprised to see such different results reported in Line 226 and 233. Looking at the R Markdown code to understand how these two values have been calculated I see that calculation of VE contains the test predictions within the denominator in:

validation[i, 3] <- 1-(mse(df$test.SedRate, df$test.pred)/var(df$test.SedRate, df$test.pred))

As VE is calculated as the unexplained variation over the total variation its not clear to me why the denominator in your calculation has the test set predictions. Suggest checking your formulas to ensure the values presented are correct. Its also not clear to me whether both metrics are required or tell a story that is not captured by r^2. So you may wish to present r^2 only.

Reply: We acknowledge that the above R code line is indeed erroneous. We also agree that it is sufficient to use either $r^2$ or variance explained. The code was corrected, and relevant text (methods and results) updated accordingly.

Line 253: Starting the discussion by referring to the R Markdown code and seemed a little out of place. As the results of this study are a valuable contribution to the field of Blue Carbon which is rapidly gaining interest to develop policies in various European governments, this focus on encouraging use of the scripts may be less interesting to the reader. Authors may consider moving this to section '6.4 Suggestions for future research' and instead focussing on the main findings of the study.

Reply: We agree that the beginning of the discussion could be improved. The first paragraph of the discussion now reads:

We have presented estimates of OC stocks and accumulation rates and their associated spatially explicit uncertainties that were derived with the same modelling framework. *Our results show that a substantial amount of OC, 231 Tg within the upper 0.1 m of seabed sediment, is stored in surface sediments of the North Sea and Skagerrak. OC accumulation is effectively restricted to the Norwegian Trough, which accumulates 1.3 Tg C annually. In the following we discuss the relevance of our results*

*by comparing them with other estimates of OC stored in shelf sea sediments, coastal vegetated habitats, and terrestrial soils, which have been highlighted as significant OC stores. We further discuss zones of OC processing at the seafloor based on our regionalisation, potential implications for marine management and suggestions for future research.*

Line 259: Similar to above comment. Section '6.1 Relevance' starts with a recap of other research and not the findings of this current study. Authors should consider whether to lead with what this study has shown and then put that into context of other work to show the relevance.

Reply: This section has been revised in the meantime. It now reads:

*The surface sediments of the North Sea and Skagerrak store 230.5 ± 134.5 Tg of OC. This compares with 9.6 to 25.0 Pg C stored globally in bioturbated Holocene shelf sediments (0 – 10 cm) as estimated by LaRowe et al. (2020). Hence, sediments in the North Sea and Skagerrak store approximately 0.9 – 2.4 % of the global stock in an area that accounts for ≈1.7 % of the global shelf.*

Line 260: Does Harris et al 2014 need to be referenced here? Suggest deleting. Also, Lee et al. (2019) present maps of uncertainty for their estimates of OC. Relative to the assumptions presented in line 263-265 (total stocks vary between 12.1-24.2 Pg C), should the Lee et al. uncertainty be accounted for in this estimation, or are they at a much smaller magnitude? As the uncertainty map in Lee et al does seem to show that uncertainty is also concentrated around the continental shelf.

Reply: No longer relevant due to changes made as outlined in previous reply.

Line 264: 'between 0% and 100%'. I am struggling to follow what is being said in this sentence. Are you simply stating that the OC in 5-10cm does not exceed that in 0-5cm? As that was already stated in the previous two sentences. Unless i am missing some subtle difference.

Reply: No longer relevant, as section has been simplified (see above).

Line 284: Is this sentences stating that the shelf sediments of the European Continental Shelf are an order of magnitude greater than coastal habitats, based solely on the calculations for the North Sea/Skagerrak? Or that is the reference to 'smaller area' comparing the area covered by the North Sea/Skagerrak relative to the area covered by coastal habitats? I assume the first as no area figures have been presented for comparison of the latter. Consider rephrasing for clarity.

Reply: The sentence was rephrased:

*This indicates that shelf sediment stocks (230.5 Tg) are approximately an order of magnitude larger despite lower OC densities of 1.1 to 13.6 kg m⁻³.*

References

International Hydrographic Organization: Limits of oceans and seas, IHO Spec. Publ., 28, 39, 1953.

LaRowe, D. E., Arndt, S., Bradley, J. A., Burwicz, E., Dale, A. W. and Amend, J. P.: Organic carbon and

microbial activity in marine sediments on a global scale throughout the Quaternary, Geochim. Cosmochim. Acta, 286, 227–247, doi:https://doi.org/10.1016/j.gca.2020.07.017, 2020.

---

## Referee Report (RR1)

**Interactive Referee Comment on "Organic carbon densities and accumulation rates in surface sediments of the North Sea and Skagerrak"**

**1. General Comments**

The authors have responded fully and with great clarity to all previous review comments. I would like to thank the authors for engaging so thoroughly. The predictions (and associated uncertainties) for OCAR and sedimentation rates will provide a very useful resource for carbon cycling and budget studies. The study highlights the important role of the Norwegian tough in in accumulating and storing disproportionate amounts of OC relative to area within its sediments and clearly stands out from the North Sea areas investigated in this regard. The uncertainty estimates will provide a focus for further sampling and research efforts to plug gaps where they exist. The authors pose some interesting questions regarding zoning according to C processing and ideas for further work, including the need for a comprehensive and coordinated sampling and data collection effort to improve sedimentary carbon research. The figures are well presented and the R Markdown script a very useful output.

**2. Specific Comments**

No outstanding issues.

**3. Technical Corrections**

Minor editing suggestions as follows:

Line 8 - Suggest re-wording; Sediments don't "protect the seabed" from disturbance (I believe the authors had planned to change this previously from their response).

Line 141 – OC "content" rather than concentration if reported as weight %.

Line 315 – Suggest "as follows" instead of "in the following"?

Line 322 - 326 – Could the authors make clearer within the text that the marine stocks calculated for the different countries are lower because they only partially cover the full extent of the EEZs? Maybe this could be done by somehow merging lines 322 and 325? E.g. only 0.5% of France's EEZ falls within the extent of this study. Or perhaps, this could just be made clearer in Table 2: Caption – e.g. "Topsoil OC stocks refer to the entire area of the respective country bordering on the North Sea and Skagerrak while marine OC stocks refer to the proportion of the EEZ that falls within our study area"?

Figure 1A – Could the authors briefly explain the choice of numeric and colour scale for the Folk classes? For instance, what is the relevance of '20' and '110' having the same colour?

---

## Author Response (AR2)

**Reply to referee 1:**

**Technical corrections:**
Line 8 - Suggest re-wording; Sediments don't "protect the seabed" from disturbance (I believe the authors had planned to change this previously from their response).
Reply: This was an oversight on our behalf.
Action: We replaced "e.g., by protecting the seafloor against human-induced disturbance" with "e.g., by storing organic carbon if left undisturbed from anthropogenic activity."

Line 141 – OC "content" rather than concentration if reported as weight %.
Reply: Agreed.
Action: As suggested.

Line 315 – Suggest "as follows" instead of "in the following"?
Reply: Agreed.
Action: As suggested.

Line 322 - 326 – Could the authors make clearer within the text that the marine stocks calculated for the different countries are lower because they only partially cover the full extent of the EEZs? Maybe this could be done by somehow merging lines 322 and 325? E.g. only 0.5% of France's EEZ falls within the extent of this study. Or perhaps, this could just be made clearer in Table 2: Caption – e.g. "Topsoil OC stocks refer to the entire area of the respective country bordering on the North Sea and Skagerrak while marine OC stocks refer to the proportion of the EEZ that falls within our study area"?
Reply: Agreed.
Action: A sentence was added: "Note that topsoil OC stocks refer to the entire area of the respective country, while marine OC stocks refer to the proportion of the EEZ that falls within our study area." Additionally, the table caption was altered as suggested by the reviewer.

Figure 1A – Could the authors briefly explain the choice of numeric and colour scale for the Folk classes? For instance, what is the relevance of '20' and '110' having the same colour?
Reply: Classes '20' and '110' accidentally got the same colour. This was not intended. The numbers are internal codes for sediment classes.
Action: The colour of class '20' was changed to make it distinct from class '110'.

**Reply to associate editor:**
 - Sedimentation rate is sometimes used for deposition (of particles) rate and sometimes for sediment accumulation rate. You have used it in the latter meaning. Please mention once that you use the term sedimentation rate as short-hand for sediment accumulation rate
Reply: Agreed.
Action: First sentence in chapter 3.1.1 changed to "Linear sedimentation rate ($\omega$), measured in (cm yr$^{-1}$), is used here synonymously with sediment accumulation rate."

- Sometimes you use Tg y-1, sometimes Tg C y-1. Please be consistent and use Tg C all through
Reply: Agreed.
Action: Changes made throughout, also in the case of stocks now reported as Tg C or Pg C. One exception was, however, made when the unit Tg was followed by "of OC" (first sentence in chapter 6.1).

-line 49: replace on the other hand with however, because there is no on the one hand.
Reply: Agreed.
Action: As suggested.